# Modified Frank Wolfe in Probability Space

**Carson Kent**
Stanford University
crkent@stanford.edu

**Jiajin Li**
The Chinese University of Hong Kong
gerrili1996@gmail.com

**José Blanchet**
Stanford University
jose.blanchet@stanford.edu

**Peter Glynn**
Stanford University
glynn@stanford.edu

## Abstract

We propose a novel Frank-Wolfe (FW) procedure for the optimization of infinite-dimensional functionals of probability measures - a task which arises naturally in a wide range of areas including statistical learning (e.g. variational inference) and artificial intelligence (e.g. generative adversarial networks). Our FW procedure takes advantage of Wasserstein gradient flows and strong duality results recently developed in Distributionally Robust Optimization so that gradient steps (in the Wasserstein space) can be efficiently computed using finite-dimensional, convex optimization methods. We show how to choose the step sizes in order to guarantee exponentially fast iteration convergence, under mild assumptions on the functional to optimize. We apply our algorithm to a range of functionals arising from applications in nonparametric estimation.

## 1 Introduction

Problems in artificial intelligence, statistics, and optimization often find a common root as an infinite dimensional optimization problem in the form

$$\inf \left\{ J\left(\mu\right) : \mu \in \mathcal{P}\left(\mathbb{R}^d\right) \right\}, \tag{1}$$

for the space $\mathcal{P}\left(\mathbb{R}^d\right)$ of Borel probability measures over $\mathbb{R}^d$. In recent years, quantitative statistical and algorithmic treatments of these formulations have produced insights into modern computational methods– resulting in novel approaches to difficult, open problems. Recent works in robust optimization [6, 44, 54, 55], probabilistic fairness [57, 53], reinforcement learning [63, 64], and generative adversarial networks [42, 18, 19] highlight these gains and are linked by the following theme: problems in the form of (1) provide access to rich infinite dimensional structure that sidesteps brittle artifacts of finite dimensional formulations.

This paper provides the construction and analysis of a modified Frank-Wolfe algorithm for (1) that operates from this infinite dimensional perspective and yields concrete convergence and complexity guarantees for a sub-family of problems (1) which are well-behaved with respect to the Wasserstein distance of order 2 (see Algorithm 1 and Theorem 1). Under canonical conditions of smoothness and convexity we recover linear rates of convergence while, even for functionals which exhibit low degrees of smoothness and for conditions that go beyond convexity, we recover sublinear rates that are to be expected from finite dimensional analogues [36] (see Section 2.2).

The vanilla Frank-Wolfe method cannot work in probability space, in general, since the planar derivative (i.e. the first variation also known as the influence function) can be unbounded when distributions do not have compact support. To overcome this issue, we conduct a natural modification

35th Conference on Neural Information Processing Systems (NeurIPS 2021).

to introduce a tractable, local, linear constraint– inspired by efforts in DRO [44, 6, 25, 54]. Specifically, the modified Frank-Wolfe step admits the prototypical DRO formulation as below,

$$\sup \left\{ \int f \, d\mu : D_c(\mu, \mu_0) \leq \delta \right\}, \tag{2}$$

where $D_c(\mu, \mu_0)$ is the optimal transport cost between $\mu$ and $\mu_0$ (a reference measure) under some cost function $c$. The form of (2), itself, immediately suggests the basis of an infinite dimensional Frank-Wolfe procedure since it provides a "linear" objective subject to a local, "trust-region" constraint– centered at $\mu_0$. The relevance of (2) in distributionally robust optimization (DRO) and mathematical finance has resulted in a multitude of computational schemes [44, 34, 37, 54, 61] for solving (2). However, hitherto, such works have failed to consider (2) within the context of a general variational method for (1). What makes these efforts notable, within the scope of this work, is that they emphasize that the solution of (2) can be highly non-trivial. Indeed, without particular assumptions, (2) can disguise an computationally hard problem– despite being convex in a Banach sense on $\mathcal{P}(\mathbb{R}^d)$. Even in the case where the cost is the squared Euclidean norm $c(x,y) = \|x - y\|^2$ (the case of primary concern for this work), computational trouble can lie dormant– an artifact of inherently difficult problems in unconstrained optimization [14]. This should not be surprising, however, given specters of computational hardness dating back to early formulations of DRO [22]. To resolve these issues, we also provide a novel analysis of techniques from distributionally robust optimization (DRO) which illustrate how such formulations can be used, in a computationally tractable way, to construct first-order, variational methods. By localizing the problem to a Wasserstein ball, the new Local Linear Minimization Oracle (LLMO) that we make in this paper not only makes the vanilla Frank-Wolfe problem sensible, but it also renders the problem computationally tractable via finite dimensional convex optimization.

## 1.1 Previous work

**Distributionally robust optimization**   To navigate such pitfalls, one can consider particular instances of (2) where the objective and constraints are sufficiently structured to preclude computational intractability and permit solution via methods adapted to the provided structure. Early work with this line [27, 22, 60], has recently been supplemented by approaches [13, 26, 7, 46, 37, 67, 35] which focus directly on DRO formulations from particular contexts in machine learning and operations research. Unfortunately, the techniques offered by these efforts require assumptions which are too restrictive for this work. These assumptions typically relate to a specific form for the objective function or constraints in (2) (e.g. linear/convex functions/piecewise-convex objectives or constraints with support or density requirements, see [31, 66, 44, 34, 65, 4, 58] for additional examples). In this instance, such limitations preclude their applicability since, in general, a "gradient object" for a functional $J$ (see Section 2) need not satisfy these conditions.

A second, more relevant, approach to compute DRO problems (2) is to restrict the level of robustness $\delta$ for which the problem is solved. This technique has been used in works such as [6, 54] and we apply this principle in a similar spirit to [54]. In that work, smoothness of the objective in (2) is used to, qualitatively, argue that a sufficiently small $\delta$ provides a computationally-tractable optimization problem. In contrast, we provide quantification of the level of robustness required to achieve this tractability and we demonstrate that this level robustness is sufficiently large to achieve canonical convergence rates for an infinite-dimensional Frank-Wolfe algorithm.

**Variational methods**   Although formulation of a Frank-Wolfe method for (1) (with quantitative bounds on complexity and convergence) has not appeared in previous literature, certain, tangentially-related variational methods offer conceptual similarity. The first of these methods is [41], which draws similar inspiration from finite-dimensional Frank-Wolfe procedures. However, the notion of first-order variation in [41] appears to be induced by the total variation distance– requiring compact support assumptions for the problem. Alternatively, we exploit Wasserstein geometry to provide a weaker notion of first-order variation, namely, Wasserstein differentiability– allowing us to eliminate restrictions to compact support. Hence, by balancing tractability of the analysis and fidelity of the procedure, the proposed algorithm scheme can be applied a broad class of computational examples. Indeed, [41] only discussed the Sinkhorn barycenter problem and it is still unclear how it can be applied to our setting. The second, more closely related effort, is [39] where similar, infinite-dimensional conditions (Section 2.2) are used to study particle-based methods for computing Nash equilibria of zero-sum games. While the setting and procedures developed in this work are completely

different, we note that, as a special case, our Frank-Wolfe method can also produce a particle-based optimization procedure. This hints at possible insightful connections with other particle methods [40, 24, 11, 10] which are left for future consideration.

## 2 Wasserstein geometry

This work considers the problem

$$\min_{\nu \in \mathcal{P}_2(\mathbb{R}^d)} J(\nu) \tag{3}$$

for functionals $J : \mathcal{P}_2(\mathbb{R}^d) \to \bar{\mathbb{R}}$ over the subset of Borel measures $\mathcal{P}(\mathbb{R}^d)$ defined by

$$\mathcal{P}_2(\mathbb{R}^d) := \left\{ \mu \in \mathcal{P}(\mathbb{R}^d) : \int_{\mathbb{R}^d} \|x\|^2 \, d\mu(x) < \infty \right\} \tag{4}$$

In particular, we consider functionals $J$ that are differentiable (Definition 1) with respect to Wasserstein distance of order 2

$$\mathcal{W}^2(\mu, \nu) := \inf_{\gamma \in \Pi(\mu, \nu)} \int_{\mathbb{R}^d \times \mathbb{R}^d} \|x - y\|^2 \, d\gamma(x, y) \tag{5}$$

where $\Pi(\mu, \nu)$ is the set of all joint couplings with marginals $\mu, \nu \in \mathcal{P}_2(\mathbb{R}^d)$. Common examples of functionals that can be cast within this framework are as follows.

**Example 1** (Divergences). For any convex, lower-semicontinuous function $f : \mathbb{R}_+ \to \mathbb{R}$ such that $f(1) = 0$, one can consider a "$f$-divergence" of the form

$$J(\mu) := D_f(\mu\|\nu) = \int_{\mathbb{R}^d} f\left(\frac{d\mu}{d\nu}\right) \, d\nu. \tag{6}$$

For instance, if $f(t) = t \log t$, (6) reduces to Kullback-Leibler divergence $D_{KL}$.

**Example 2** (Integral Probability Metrics). For a set of real valued functions $F$ on $\mathbb{R}^d$ one can define the discrepancy

$$J(\mu) := \text{IPM}(\mu, \nu) = \sup_{f \in F} \int_{\mathbb{R}^d} f \, d\mu - \int_{\mathbb{R}^d} f \, d\nu \tag{7}$$

for $\mu, \nu \in \mathcal{P}(\mathbb{R}^d)$, where $\nu$ is a fixed, reference measure. Such discrepancies are termed Integral Probability Metrics (IPMs), although they may not strictly satisfy the requirements of a metric– say, by failing to distinguish all pairs of measures. Instead, for a pair of measures $\mu, \nu \in \mathcal{P}(\mathbb{R}^d)$, IPMs can be interpreted as measuring the extent to which $\mu$ and $\nu$ differ on functions in $F$– or, rather, measuring the extent to which $\mu$ and $\nu$ can be distinguished by $F$. Concretely, consider $F = \{f : \|f\|_H \leq 1\}$ where $H$ is a reproducing kernel hilbert space (RKHS). In this case, one obtains the dual formulation of Maximum Mean Discrepancy (MMD).

### 2.1 Properties and differentiability for Wasserstein space

Under the Wasserstein distance, $\mathcal{P}_2(\mathbb{R}^d)$ is a Polish space [59] and, more importantly, it is a *geodesic* space. That is, for every $\mu, \nu \in \mathcal{P}_2(\mathbb{R}^d)$, there exists a constant-speed geodesic curve $\mu_t : [0, 1] \to \mathcal{P}_2(\mathbb{R}^d)$ where $\mu_0 = \mu$, $\mu_1 = \nu$ and

$$\mathcal{W}(\mu_t, \mu_s) = |t - s|\mathcal{W}(\mu_0, \mu_1) \tag{8}$$

Moreover, there is a bijection between constant-speed geodesics and optimal transport plans [1, Theorem 7.2.2]. Every geodesic corresponds to a unique, optimal transport plan $\gamma \in \Pi(\mu, \nu)$ such that

$$\mu_t = ((1 - t)x + ty)_\# \gamma \qquad \text{where} \quad \mathcal{W}^2(\mu, \nu) = \int_{\mathbb{R}^d \times \mathbb{R}^d} \|x - y\|^2 \, d\gamma(x, y), \tag{9}$$

and $\mu_t$ is the distribution of the random variable $(1 - t)X + tY$ with the pair $(X, Y)$ following distribution $\gamma$. Conversely, every optimal transport plan gives rise to a unique geodesic via (9).

Since our Frank-Wolfe method minimizes a sequence of linear approximations, one must define the notion of a gradient (of a functional $J$) to be compatible with respect to the geometry of these geodesics. In particular, as $\mathcal{P}_2(\mathbb{R}^d)$ is curved under $\mathcal{W}$ (see Appendix A), gradients must be defined in terms of a selection in an appropriate cotangent bundle. For Wasserstein space, this cotangent bundle (denoted $\text{CoTan}_{\mathcal{P}_2(\mathbb{R}^d)}$) is essentially the set of vector fields on $\mathbb{R}^d$ that can be approximated by gradients of smooth functions (see Appendix A for details). This results in the following definition.

**Definition 1** (Wasserstein differentiability)**.** Let $S$ be a geodesically convex set; that is, $\mu_t \in S$ for any geodesic $\mu_t$ between $\mu, \nu \in S$. A functional $J : \mathcal{P}_2(\mathbb{R}^d) \to \mathbb{R}$ is Wasserstein differentiable on $S$ if there is a map $F : \mathcal{P}_2(\mathbb{R}^d) \to \mathrm{CoTan}_{\mathcal{P}_2(\mathbb{R}^d)}$ such that for all $\mu, \nu \in S$ and a geodesic $\mu_t : [0, 1] \to \mathcal{P}_2 (\mathbb{R}^d)$ between $\mu$ and $\nu$, one has

$$\lim_{\alpha \to 0^+} \frac{J(\mu_\alpha) - J(\mu)}{\alpha} = \int_{\mathbb{R}^d \times \mathbb{R}^d} F(\mu; x)^T (y - x) \, d\gamma(x, y), \tag{10}$$

where $\gamma$ is the unique optimal transport plan (9) corresponding to $\mu_t$. Note that $F(\mu; x) = (F(\mu))(x)$ provides an aesthetic way of representing the evaluation at $x \in \mathbb{R}^d$ of the output of $F$ at $\mu$. The map $F$ is called the Wasserstein derivative of $J$.

*Remark* 1. The description of differentiability provided by Definition 1 falls within the general framework of metric derivatives and Wasserstein gradient flows, largely codified in [1]. This framework is now a well-established component of the theory of Wasserstein spaces, while the relation (10), itself, presents only a narrow structuring of ideas from this framework. Definition 1, however, is often how works in statistical and algorithmic fields interact with this broader area [56, 16, 38, 39]. Moreover, this literature demonstrates the most motivating feature of (10): a large number of functionals of interest for machine learning and statistical inference exhibit Wasserstein gradients in the sense of (10). The curious reader is referred to [1, 52, 9] for precise statements of conditions under which (10) is guaranteed. However, $F$ is intimately relative to the Gateaux differential for $J$ [52, 56]. Recall, the Gateaux differential for a functional $J$ exists when there is an appropriate, dual space $D^*$ on a closed subspace $D \subseteq \mathcal{P}(\mathbb{R}^d)$ such that

$$\langle dJ(\mu), \nu - \mu \rangle = \lim_{\alpha \to 0^+} \frac{J(\mu + \alpha(\nu - \mu)) - J(\mu)}{\alpha} \tag{11}$$

for some $dJ(\mu) \in D^*$ and all $\mu$ in some set $S$ such that $S - S \subseteq D$. In instances where the Gauteaux differential $dJ(\mu)$ exists, the Wasserstein derivative $F$ will usually exist [39] and be given by $\nabla dJ(\mu)$. Here, we use the finite dimensional gradient operator $\nabla(\cdot)$ formally, and omit a rigorous exposition on this operation in the context of $\mathrm{CoTan}_{\mathcal{P}_2(\mathbb{R}^d)}$.

It should be noted that computation of the Wasserstein derivative might be difficult. Indeed, for a $J$ in a variational form such as (7), computation of the Wasserstein derivative is equivalent to finding a *witness* function that achieves the supremum [52]. In the case of a pathological sets (in (7)), such a task might be intractable. To resolve this issue, and to simplify our treatment, this work utilizes the existence of an oracle for the computation of a Wasserstein gradient. This oracle permits a unified description of our Frank-Wolfe algorithm and abstracts away variation in functional-specific computational cost. Recall that a function in the Hölder space $C^1(\mathbb{R}^d)$ is called $L$-smooth if has $L$-Lipschitz gradients.

**Definition 2** (Wasserstein Derivative Oracle)**.** Let $J : \mathcal{P}_2(\mathbb{R}^d) \to \mathbb{R}$ be a Wasserstein differentiable functional on a set $S$ with Wasserstein derivative $F : \mathcal{P}_2(\mathbb{R}^d) \to \mathrm{CoTan}_{\mathcal{P}_2(\mathbb{R}^d)}$. A $L$-smooth Wasserstein derivative oracle over $S$ is an oracle which, given sample access to a distribution $\mu \in S$ and an error parameter $\epsilon$, returns an $L$-smooth function $\widehat{\phi}_\mu \in C^1(\mathbb{R}^d)$ satisfying

$$\left\| \nabla \widehat{\phi}_\mu - F(\mu) \right\|_{L^2(\mu)} \leq \epsilon \tag{12}$$

where $\|\cdot\|_{L^2(\mu)}$ is the canonical norm on the space $L^2(\mu)$ of square integrable functions with respect to $\mu \in \mathcal{P}(\mathbb{R}^d)$. In this work, the output of this oracle is represented as $\Theta(\mu, \epsilon)$.

*Remark* 2. The qualification that the Wasserstein derivative oracle return an $L$-smooth function is necessary to exclude the, aforementioned, possibility of a pathological Wasserstein derivative– that would be intractable for use in a computational procedure. In some ways, this is representative of the fact that the cotangent space $\mathrm{CoTan}(\mu)$ at a point $\mu$ is too large. Such a condition is common in other variational methods [3, 19, 64, 20] and is relatively superficial– when coupled with the degree of approximation afforded by $\epsilon$. Via smoothing techniques [51, 11, 38], functionals can often be assumed to have Wasserstein derivatives which are $C^1(\mathbb{R}^d)$ or are well-approximable by $C^1(\mathbb{R}^d)$ functions.

## 2.2 Smoothness and Łojasiewicz inequalities

In finite dimensions, iterative, gradient-based methods typically require the specification of two conditions in order to achieve convergence.

- The accuracy of local, linear approximations that are provided by the gradient.
- The extent to which local descent makes global progress on the objective.

Here, we state these conditions in the context of functionals over Wasserstein space.

**Definition 3** ($\alpha$-Hölder smoothness). Let $S$ be a geodesically convex set and let $J : \mathcal{P}_2\left(\mathbb{R}^d\right) \to \mathbb{R}$ be a functional which is continuously Wasserstein differentiable on the set $S$. $J$ is said to be locally $\alpha$-Holder smooth on $S$ with parameters $T$ and $\Delta$ if for all $\mu \in S$ and all $\nu \in S$ such that $\mathcal{W}(\mu, \nu) \leq \Delta$, there exists an optimal transport plan $\gamma \in \mathcal{P}_2(\mathbb{R}^d \times \mathbb{R}^d)$ such that

$$J(\nu) \leq J(\mu) + \int_{\mathbb{R}^d \times \mathbb{R}^d} F(\mu; x)^T (y - x) \, d\gamma(x, y) + \frac{T}{1 + \alpha} \mathcal{W}^{1+\alpha}(\nu, \mu) \tag{13}$$

**Definition 4** (Łojasiewicz inequality). A Wasserstein differentiable functional $J$ on a set $S \subseteq \mathcal{P}_2(\mathbb{R}^d)$ is said to satisfy a *Łojasiewicz inequality* with parameter $\tau$ and exponent $\theta$ if for all $\mu \in S$ and $J_* := \inf_{\mu \in S} J(\mu)$

$$\tau \left( J(\mu) - J_* \right)^\theta \leq \|F(\mu)\|_{L^2(\mu)} \tag{14}$$

where $F$ is the Wasserstein derivative (10) of $J$.

*Remark* 3. More restrictive versions of both (13) and (14) commonly appear in previous literature to establish the explicit converence rate [3, 32, 39, 19, 15]. In most cases, the $\alpha$-Hölder smoothness condition (13) is stated for $\alpha = 1$ and required to hold globally ($\Delta = \infty$). This smoothness criterion is considerably weaker since it requires that the Wasserstein gradient only provide a local approximation that is slightly more than first-order accurate. Moreover, such a condition can be necessary when the Wasserstein derivative (10) is not Lipschitz with respect to $\mathcal{W}$ in the cotangent space norm on $\text{CoTan}_{\mathcal{P}_2(\mathbb{R}^d)}$– see [43] for such an example. Additionally, statement of the Łojasiewicz inequality (14) is broader than canonical treatments due to the presence of the auxiliary power $\theta$. Most often, the specific instances of either $\theta = 1/2$ or $\theta = 1$ are considered. The case $\theta = 1$ is implied for (geodesically) convex functionals $J$ with a $\mathcal{W}$-bounded level set, while $\theta = 1/2$ is implied for strongly convex $J$ [1, 32]. Although the notions of Holder smoothness condition (14) and Łojasiewicz inequalities (13) have been well-studied in the literature [33, 5], the use of both of these conditions with explicitly determined exponents $\alpha$ and $\theta$ to provide concrete convergence rates for a computationally-implemented, infinite-dimensional descent method does not appear in related literature as far as the authors are aware.

## 3 Modified Frank-Wolfe algorithm

Algorithm 1 provides our modified Frank-Wolfe procedure along with its associated convergence guarantees and sample complexities in Theorem 1 and Proposition 1. It is worth mentioning that the algorithm itself only requires a much weaker notion of differentiability to be applicable, namely, Gateaux differentiability (i.e., $dJ(\mu)$ in (11) exists). In Section 4, we will provide several computational examples which may not satisfy the following assumptions but work well in practice. We have to admit there is a theoretical and computational gap. We will leave it as an open question for our future work. However, to conduct the convergence analysis, we require the following assumptions–phrased in the language from Section 2.

**Assumption 1** (Smoothness assumption). The functional $J : \mathcal{P}_2(\mathbb{R}^d) \to \bar{\mathbb{R}}$ is Wasserstein differentiable (Definition 1) and locally $\alpha$-*Holder smooth* (Definition 3) on a set $S \subseteq \mathcal{P}_2(\mathbb{R}^d)$ with parameters $T$ and $\Delta_1 > 0$ (Definition 3). Further, an $L$-smooth Wasserstein derivative oracle (Definition 2) for $J$ exists.

**Assumption 2** (Local richness). The set $S$ is rich enough to contain the solution to (2) for $\mu \in S$, $L$-smooth $-f$, and $\delta \leq \Delta_2$.

*Remark* 4. When $S$ is not the whole set, it is necessary to invoke Assumption 2 to guarantee that the iterates produced by our algorithm remain in $S$. This is a result of the fact that the solution of (15) is optimal for some Wasserstein ball of size $\tilde{\delta} \leq \delta$. Hence, each of these iterates is guaranteed to lie in $S$ so long as $\delta \leq \Delta_2$.

**Assumption 3** (Łojasiewicz assumption). The functional $J$ satisfies a *Łojasiewicz inequality* (14) on $S \subseteq \mathcal{P}_2(\mathbb{R}^d)$ with parameters $\tau > 0$ and $\theta$.

---

**Algorithm 1** Modified Frank Wolfe for (3)

---

**Input:** Wasserstein derivative oracle $\Theta$, initial distribution $\mu_0$, smoothness parameter $\alpha$, gradient error $\hat{\epsilon}$, estimation error $\bar{\epsilon}$, iterate error $\widetilde{\epsilon}$, stopping threshold $r$, step sizes $(\beta_1, \beta_2, \beta_3)$, number of iterations $k$

    **for** $1 \leq i \leq k$ **do**

        Let $\widehat{\phi}_{\mu_{i-1}} \leftarrow \Theta(\mu_{i-1}, \hat{\epsilon})$                                       ($\triangleright$) Definition 2

        Compute $\left\| \nabla \widehat{\phi}_{\mu_{i-1}} \right\|_{L^2(\mu_{i-1})} - \bar{\epsilon} \leq s \leq \left\| \nabla \widehat{\phi}_{\mu_{i-1}} \right\|_{L^2(\mu_{i-1})}$

        **if** $s \leq r$, **then break**

        **else**   $\delta \leftarrow \min\left(\beta_1, \beta_2 s, \beta_3 s^{1/\alpha}\right)$, $\zeta \leftarrow \delta\widetilde{\epsilon}$

        Compute $\mu_i$ satisfying $W(\mu_i, \mu_{i-1}) \leq \delta$ and               ($\triangleright$) Proposition 1

$$\int \widehat{\phi}_{\mu_{i-1}} \, d\mu_i - \inf_{\mathcal{W}(\nu, \mu_{i-1}) \leq \delta} \int \widehat{\phi}_{\mu_{i-1}} \, d\nu \leq \zeta \tag{15}$$

    **return** $\mu_i$

---

**Theorem 1.** *Under Assumptions 1, 2, 3, and an appropriate choice of input parameters, Algorithm 1 computes a distribution $\mu^*$ satisfying*

$$r(\mu^*) := J(\mu^*) - \inf_{\mu \in S} J(\mu) \leq \epsilon \tag{16}$$

*in at most*

$$k = \widetilde{O}\left(\epsilon^{-p_-}\right) \tag{17}$$

*iterations, where $p_-$ denotes the negative part of $p = 1 - \alpha^* \theta$ for the dual exponent $\alpha^* = (1 + \alpha)/\alpha$. The notation $\widetilde{O}(\cdot)$ omits logarithmic factors in it's arguments.*

*Remark* 5. In the case of a (geodescially) strongly-convex and 1-Hölder smooth functional $J$, the Łojasiewicz inequality (14) holds with $\theta = 1/2$ and one obtains standard $\widetilde{O}(1)$ complexity (in terms of $\epsilon$). This is to be expected from finite dimensional analogues [36]. Similarly, for $J$ which is only convex (with a $\mathcal{W}$-bounded level set), (14) holds with $\theta = 1$ and (17) yields a $\widetilde{O}(\epsilon^{-1})$ complexity that mimics canonical results. The step size required to achieve these complexities is illustrated by the choice of $\delta$ in Algorithm 1.

For functionals which are $\alpha$-Hölder smooth for $\alpha < 1$, the dependence on the dual exponent $\alpha^*$ in (17) can be rather punishing for small $\alpha$. It is natural to ask if this exponent could be improved within the scope of Assumptions 1, 2, and 3. Moreover, in finite dimensions, it is well known that first-order methods for convex and $\alpha$-Hölder smooth functions (also known as weakly smooth functions) can obtain $\epsilon$-optimal solutions in $O(\epsilon^{-2/(1+3\alpha)})$ iterations [47]. Hence, it could even be considered whether, given geodesic-convexity assumptions on $J$, a better iteration complexity for Algorithm 1 would be obtainable.

We conjecture that such improvements are unlikely - particularly those that would draw on analogy from finite dimensional techniques. The motivation for this is as follows. A common approach to establishing improved iteration complexities for convex, $\alpha$-Hölder smooth functions, in finite dimensions is to consider their gradient oracles as inexact oracles for convex, 1-Hölder smooth functions [23]. Using either averaging arguments or accelerated methods, more rapid progress on an underlying objective can then be made with these inexact oracles. Our Frank-Wolfe method already utilizes an inexact step (15) so, conceptually, such an approach could be applied to Algorithm 1. Unfortunately, however, such finite dimensional analogies fail due to the difficulty of averaging distributions in Wasserstein space. Indeed, in finite dimensions, averaging is crucial to prevent error accumulation from outpacing objective progress. Since Wasserstein space is positively curved (Appendix A) computing analogous convex combinations of the $\mu_i$ in Algorithm 1 is, itself, a variational problem that might be as expensive to compute as the original problem (3).

### 3.1 Computation of the Frank-Wolfe step

An algorithm for computation of the Frank Wolfe step (15) is given in Appendix C and the net result of this procedure is as follows.

**Proposition 1.** *Under the assumptions of Theorem 1, there exists a stochastic algorithm which (with high probability) provides sample access to a distribution satisfying* (15). *Moreover, in the setting of* (17), *this algorithm requires* $\widetilde{O}\left(\epsilon^{-2\alpha^*\theta}\right)$ *samples and gradient evaluations.*

Crucially, the property enabling Proposition 1 is *duality*. The Frank-Wolfe problem (15) exhibits a dual of the form

$$\sup_{\lambda\geq 0}\quad \left[\int_{\mathbb{R}^d}\left(\inf_{y\in\mathbb{R}^d}f(y)+\frac{\lambda}{2}\left\|x-y\right\|^2\right)d\mu(x)-\frac{\delta^2\lambda}{2}\right] \tag{18}$$

and (18) is amenable to computation using techniques from finite dimensional optimization. This approach derives from distributionally robust optimization [6, 54, 25, 34] where such techniques have been used to produce methods in optimization and machine learning that are robust to adversarial perturbations.

In general, solution of (15) for any level of $\delta$ could be computationally hard [6, 54]. However, since a Frank-Wolfe procedure need only solve *local* problems, not *global* ones, we show: there is a $\delta$ which is, simultaneously, small enough to enable the efficient computation of (18)), yet large enough to produce (17). The techniques used to achieve these results most closely resemble ideas from [54]. However, we provide a precise quantification of the $\delta$ in (15) that is required to achieve computation tractability and develop a procedure which yields guarantees on the primal-dual gap of (15) and (18). In turn, the theoretical insights that we obtain suggest an empirical procedure in which $\lambda$ in (18) can be updated relatively infrequently within our Frank-Wolfe algorithm, provided that it is chosen on a proper scale. Such an implementation is investigated in Section 4 and yields significant computational savings.

It is also worth noting that the implementation in Appendix C requires only sample access to $\mu_0$ in order to provide sample access to a distribution satisfying (16). Thus, all operations in Algorithm 1 can be implemented using only sample access to $\mu_0$. Practically, however, it is often more efficient to maintain approximations to the iterates $\mu_i$ via a non-parametric estimator. When this is done, it results in an additional, additive error in the residual (16) at each step of Algorithm 1. If this error is on the order of the error produced by the Wasserstein derivative oracle $\Theta$, the iteration complexity (17) remains unaffected. Moreover, analysis of the error induced by a particular non-parametric approximation of the $\mu_i$ is highly problem dependent– so we do not consider it in the context of Theorem 1.

## 4 Computational examples

In this section, we demonstrate the application our Frank-Wolfe algorithm to several non-parametric estimation problems in statistics and machine learning. All simulations are implemented using Python 3.8 on a high performance computing server running Ubuntu 18.04 with a Gen10 Quad Intel(R) Xeon(R) Platinum 8268 CPU @ 2.90GHz processor. As we mentioned in Section 3, the proposed algorithm just needs a very weak of differentiability to be applied (i.e., Gateaux differentiability). The computational examples we conduct in this section aim to show attractive performance in applications where the assumptions required for theoretical convergence are unknown, instead of corroborating our theoretical results.

### 4.1 Gaussian deconvolution

A classical task in nonparametric statistics [12, 8] is to infer a latent, data-generating distribution $\mu\in\mathcal{P}_2\left(\mathbb{R}^d\right)$ from a set of observations that are corrupted by independent, additive Gaussian noise. For observations $Y_1,\ldots,Y_n$ such that $Y_i=X_i+Z_i$ where $X_i\sim\mu,Z_i\sim N\left(0,\sigma^2\right)$. One seeks to compute a non-parametric estimate of $\mu$ — the variance of the noise $\sigma^2$ is considered known. Since $Z_i$ is independent of $X_i$, this task amounts to "deconvolving" $\mu$ from the distribution of $Z_i$. A natural candidate for $\mu$ is the maximum-likelihood estimator (MLE),

$$\widehat{\mu}:=\arg\max_{\mu\in\mathcal{P}_2(\mathbb{R}^d)}\ \sum_{i=1}^n\log\int_{\mathbb{R}^d}g_\sigma\left(Y_i-x\right)d\mu(x) \tag{19}$$

where $g_\sigma$ is the density of $Z_i$. We refer the reader to [50, Section 3] for further details but note that it was shown in [50] that $\hat{\mu}$ has an equivalent characterization

$$\hat{\mu} = \underset{\mu \in \mathcal{P}_2(\mathbb{R}^d)}{\arg\min} \quad D_{\sigma^2}(\mu, \hat{P}_Y) \tag{20}$$

where

$$D_{\sigma^2}(\mu_1, \mu_2) := \inf_{\pi \in \Pi(\mu_1, \mu_2)} \frac{1}{2} \int \|x - y\|^2 \, d\pi(x, y) + \sigma^2 D_{KL}(\pi \,\|\, \mu_1 \otimes \mu_2) \tag{21}$$

is the entropic optimal transportation distance [21] and $\hat{P}_Y$ is the empirical distribution of the $Y_i$. The problem (20) readily lies within the framework of (3) for $J(\mu) := D_{\sigma^2}(\mu, \hat{P}_Y)$. Moreover, it is known [41] that the Wasserstein derivative (10) of $D_{\sigma^2}(\mu, \hat{P}_Y)$ with respect $\mu$ is given by

$$\nabla \hat{\phi}_\mu(x) = \sigma^2 \log \left( \frac{1}{n} \sum_{i=1}^{n} \exp \left( (v_i^* - \|x - y_i\|^2/2) / \sigma^2 \right) \right) \tag{22}$$

where $v^* \in \mathbb{R}^n$ is dual variable (corresponding to $\hat{P}_Y$) which is optimal for $D_{\sigma^2}(\mu, \hat{P}_Y)$. This provides a Wasserstein derivative oracle for (20) as the vector $v^*$ can be readily approximated using sinkhorn or stochastic gradient algorithms [49].

**Toy example On 2D Gaussian mixture** A simple, two dimensional instance of (19) is shown in Figure 1 on a dataset $Y_i$ of 50 samples with mixture of 4 Gaussians — illustrated by the kernel density estimator of the $Y_i$, shown in red. The behavior of our Frank-Wolfe Algorithm is depicted over the course of several iterations, where the foreground contours provide the density of the iterate, $\mu_i$, that is maintained by the algorithm. It can be easily observed that despite the small overlap between our initial distribution and target one, our method reaches the global optima very quickly.

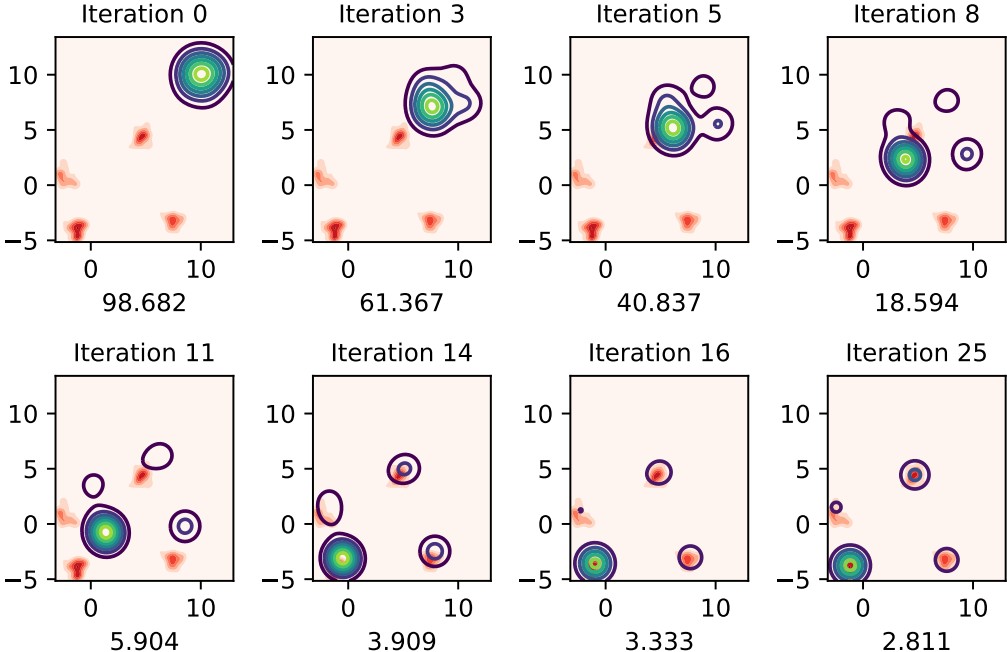

Figure 1: Toy Example on a 2D Gaussian Mixture. Note that the initial distribution is set as $N([10, 10], \sigma^2 \mathbf{I})$ for $\sigma^2 = 0.4$ and the number of particles is 200. The bisection method of Appendix C is used with tolerance set to $1e^{-3}$. The objective value $D_{\sigma^2}(\mu_i, \hat{P}_y)$ is below to each sub-figure.

**Uniform strategy for $\lambda$ and sensitivity analysis on high-dimensional examples** Since the number of bisection ascent step for $\lambda$ in (18) that arises during the search can be large, the original algorithm may be computationally rather demanding for high-dimensional cases. Thus, we are

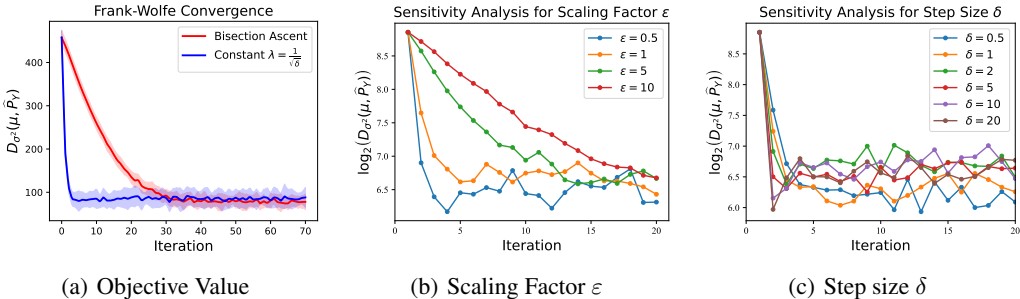

Figure 2: High-dimensional Gaussian deconvolution for $d = 64$. For 50 data points, $Y_i$, sampled from a mixture of 7-Gaussians, 200 particles are used in a non-parametric estimate the $\mu_i$. In Figure 2(a), the tolerance for the ascent method is $1e^{-3}$ and the shaded bands show the standard derivation over 10 independent runs with random initializations. In Figure 2(b), the step size $\delta$ is $0.5$.

motivated to develop an approach that makes $\lambda$ as a constant which only depends on the step size $\lambda = \varepsilon/\delta^{1/2}$ without other one-dimensional optimization methods as an inner solver, where $\varepsilon$ is the scaling factor. Figure 2(a) provides a convergence behavior comparison between the vanilla Algorithm 1 and the modification leveraging this new uniform strategy. Not surprisingly, the modified Frank-Wolfe algorithm can get to the local region around the global optima faster but with a larger variance, as the uniform strategy is indeed a more aggressive strategy at the early stage. To further support the uniform strategy and our Frank-Wolfe framework, we conduct extensive sensitivity experiments on the hyperparameter (e.g., step size $\delta$ and scaling factor $\varepsilon$). Both Figure 2(b) and 2(c) demonstrate that our algorithm is robust to these crucial hyperparamter. We can also observe that it is better not to choose a relatively large step size although we can converge faster at the beginning but suffer from the risk of divergence, as the maximum step size is controlled by the smooth parameter in theory. Hence, our experiment results here also corroborate the theoretical findings.

It is worth mentioning that this new uniform strategy can make our Frank-Wolfe framework be extended to an asynchronous decentralized parallel setting easily and thus can further meet the requirements of large-scale applications. Based on the superior performance, we left its rigorous convergence analysis as an open question.

## 4.2 Nonparametric learning with student-teacher networks

The rise of generative adversarial networks (GANs) [28] and efforts connecting neural networks and kernel regression [17], have generated interest in maximum mean discrepancy (MMD), particularly with respect to it's role in constructing high-dimensional, distributional embeddings [20, 45]. This development is predicated on the observation that any neural network $(x, \theta) \rightarrow \psi(x, \theta)$, which produces an output $\psi(x, \theta) \in \mathbb{R}^d$ from input data $x \in X \subseteq \mathbb{R}^d$ and parameters $\theta \in \Theta \subseteq \mathbb{R}^m$, yields a kernel on the parameter set $\Theta$:

$$k(\theta_1, \theta_2) := \mathbb{E}_x \left[ \psi(x, \theta_1)^T \psi(x, \theta_2) \right] \tag{23}$$

where the expectation over $x$ is taken with respect to a data generating distribution. Via MMD, $k(\cdot, \cdot)$ induces a natural discrepancy measure between distributions over network parameters $\theta$. Thus, learning of a generative image model can be expressed as minimizing MMD with respect to latent, generative distribution for $\nu$. We refer to [45, 3] for further descriptions of these applications.

Being an integral probability metric (7), squared MMD lies well within the framework of this paper

$$J(\mu) := \text{MMD}^2(\mu, \nu). \tag{24}$$

and the Gateaux derivative (i.e., influence function) of $J$ admits a natural expression [3] as the difference between the mean embeddings of $\mu$ and $\nu$

$$f_\mu^*(x) = \mathbb{E}_{z \sim \mu} \left[ k(z, x) \right] - \mathbb{E}_{z \sim \nu} \left[ k(z, x) \right] \tag{25}$$

Indeed, $f_\mu^*$ can be readily computed via sampling methods– even when $\mu$ or $\nu$ are continuous or are large, discrete distributions [29]. Note that, as discussed in Remark 1, the Wasserstein

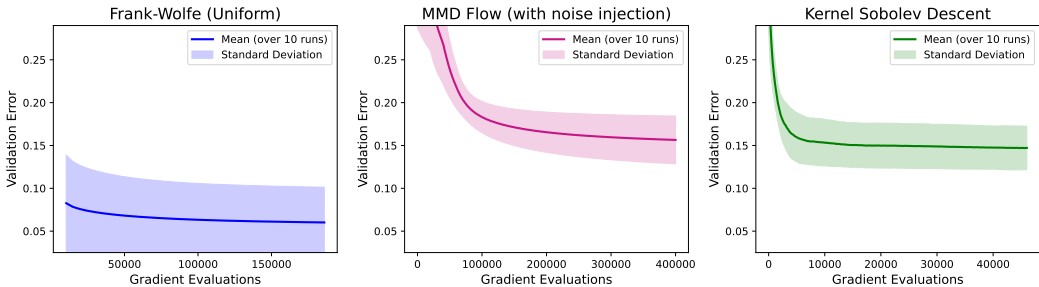

Figure 3: Student-Teacher Network; The detailed implementation set up is same as [3, Appendix G]. The left one is the result for our Frank-Wolfe method with the uniform strategy $\lambda = \frac{0.05}{\delta}$ and the step size $\delta$ is 0.5. The number of particle is 200.

derivative is, under sufficient regularity, the gradient of the Gateaux differential $\nabla f_\mu^*(\cdot)$. Perhaps the most advantageous consequence of (25), however, is that the Wasserstein gradient directly inherits regularity present in $k$. Indeed, should $\nabla_x k(x, y)$ be $L$-Lipschitz in $x$ (uniformly for all $y$), $J$ (24) is naturally $L$-smooth [3]. This has led to the development of several variational or particle-based methods for minimizing (24) [3, 45, 20].

*Remark* 6. For general MMD functionals, the smoothness and Łojasiewicz inequalities (i.e., Assumptions 1 and 3 ) are shown in [3]. Nevertheless, the MMD experiments in our paper, following the setup in [3], fail to satisfy the differentiability assumptions in [3] due to the ReLU terms present in the network defining the kernel.

However, despite a possible violation of the assumptions, Figure 3 demonstrates the competitive performance of our method with two of baselines showcased in [3] on Student-Teacher network problem. Our method is shown on the left, the center plot shows the "MMD gradient flow" algorithm from [3], and the right plot provides the "Sobolev Descent" algorithm of [45]. Performance is evaluated in terms of MMD error on a validation dataset and is shown as a function of the total gradient evaluations performed by each method. This provides a better proxy for relative performance and convergence since an iteration of Algorithm 1 performs multiple solves that are, each, similar in terms of gradient complexity to a single iteration of MMD gradient flow or Sobolev descent. Further, the total number of gradient evaluations should not be viewed as a proxy for wall-time as, for each gradient evaluation, the number of operations performed by each method can vary widely. Indeed, for each gradient evaluation in Sobolev descent an entire linear system solve is performed, which is computational demanding in practice. Also, note that, as both MMD gradient flow and Sobolev descent are particle-based, Algorithm 1 was, for the purposes of comparison, instantiated with a particle distribution of equal size.

# 5 Conclusion

This paper introduces and studies a Frank-Wolfe procedure for the minimization of functionals of probability measures. While these methods have been widely studied in the finite-dimensional setting; our current environment presents both significant benefits and opportunities. First, many problems of interest can be posed in terms of the types of formulations that we study [18, 19, 6, 54, 20, 39, 64, 55, 57, 11]. Second, our algorithm can naturally be asynchronously parallelized. This is a research avenue of significant promise that we plan to explore in future work, especially in connection with the wide range of applications mentioned earlier.

**Acknowledgements**    Material in this paper is based upon work supported by the Air Force Office of Scientific Research under award number FA9550-20-1-0397. Additional support is gratefully acknowledged from NSF grants 1915967, 1820942 and 1838576.

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
