# A  Properties of Wasserstein space

The following properties of Wasserstein space make Definition 1 precise and are using in the convergence proofs in Appendix B.

**Proposition 2** (Properties of Wasserstein space)**.**

- *For a constant-speed geodesic $\mu_t : [0,1] \to \mathcal{P}_2(\mathbb{R}^d)$ with respect to $\mathcal{W}$, there exists a ($\mu_t$-almost surely) unique Borel vector field $v_t : [0,1] \times \mathbb{R}^d \to \mathbb{R}^d$ which satisfies*

$$\mathcal{W}^2(\mu_0, \mu_1) = \int_0^1 \int_{\mathbb{R}^d} \|v_t(x)\|^2 \, d\mu_t(x) \, dt = \min_{v_t \in V_\mu} \int_0^1 \int_{\mathbb{R}^d} \|v_t(x)\|^2 \, d\mu_t(x) \qquad (26)$$

  *for*

$$V_\mu := \left\{ v_t : \frac{d\mu_t}{dt} + \nabla \cdot (v_t \mu_t) = 0 \right\} \qquad (27)$$

  *defined as the set of all Borel vector fields which solve the continuity equation for $\mu_t$. The continuity equation is understood in duality with $C_c^\infty(\mathbb{R}^d)$.*

- *For any constant-speed geodesic $\mu_t$, the corresponding optimal transport plan $\gamma \in \Pi(\mu_0, \mu_1)$ and the corresponding vector field $v_t$ (given by (26)) satisfy the relation*

$$v_t((1-t)x + ty) = y - x, \qquad \gamma\text{-almost surely} \qquad (28)$$

  *for Lebesgue-almost every $t$.*

- *The space $\mathcal{P}_2(\mathbb{R}^d)$ is positively curved under $\mathcal{W}$ and at each point $\mu \in \mathcal{P}_2(\mathbb{R}^d)$, the tangent space*

$$\mathrm{Tan}(\mu) := \overline{\{\nabla \psi \; : \; \psi \in C_c^\infty(\mathbb{R}^d)\}}^{L^2(\mu)} \qquad (29)$$

  *is the closure in $L^2(\mu)$ of the gradients of smooth functions with compact support. Via the Riesz isomorphism, $\mathrm{CoTan}(\mu) = \mathrm{Tan}(\mu)$ where $\mathrm{CoTan}(\mu)$ denotes the cotangent space. The tangent and cotangent bundles will be denoted $\mathrm{Tan}_{\mathcal{P}_2(\mathbb{R}^d)}$ and $\mathrm{CoTan}_{\mathcal{P}_2(\mathbb{R}^d)}$, respectively.*

*Proof of Proposition 2.* To verify the first bullet, we first establish the existence of such a $v_t$. Let $\mu_t$ be the constant speed geodesic and define the set of functions

$$A_\mu := \left\{ z \in L^2([0,1]) : \mathcal{W}(\mu_t, \mu_s) \le \int_s^t z(r) \, dr \;\; \forall \, 0 \le s \le t \le 1 \right\}$$

It is clear that the function $m(r) := \mathcal{W}(\mu_0, \mu_1)$ is in $A$ and satisfies

$$m = \arg\min_{z \in A} \int_0^1 z^p(r) \, dr \qquad (30)$$

for any $p \ge 1$. Hence, the metric derivative $|\mu'|$ of $\mu_t$ fulfills

$$|\mu'|(t) = d(\mu_0, \mu_1) \qquad \text{Lebesgue almost everywhere for } t \in [0,1]$$

By Theorem 8.3.1 in [1], there exists Borel vector field $v_t : [0,1] \times \mathbb{R}^d \to \mathbb{R}^d$ satisfying the continuity equation (27) such that

$$\|v_t\|_{L^2(\mu_t)} = |\mu'|(t) = \mathcal{W}(\mu_0, \mu_1) \qquad \text{Lebesgue almost everywhere for } t \in [0,1] \qquad (31)$$

Combined with (30), this implies that $v_t$ is a solution of (26). Uniqueness of $v_t$ follows directly from the second bullet.

For a constant-speed geodesic $\mu_t$ from $\mu$ to $\nu$. Theorem 2.4 in [2] gives that, for any $\sigma \in \mathcal{P}_2(\mathbb{R}^d)$,

$$\frac{d}{dt} \frac{1}{2} \mathcal{W}^2(\mu_t, \sigma) = \int \langle v_t(x), x - y \rangle \, d\bar{\gamma}(x, y) \qquad \forall \, \bar{\gamma} \in \Pi_o(\mu_t, \sigma) \qquad (32)$$

where $\Pi_o(\mu_t, \sigma) \subseteq \mathcal{P}(\mathbb{R}^d \times \mathbb{R}^d)$ is the set of optimal transport plans between $\mu_t$ and $\sigma$. Setting $\sigma = \nu$, the fact that $\mu_t$ is a geodesic implies that there is a unique optimal coupling $\gamma \in \mathcal{P}(\mathbb{R}^d \times \mathbb{R}^d)$ between $\mu$ and $\nu$ such that

$$((1-t)x + ty, y)_{\#} \gamma \in \Pi_o(\mu_t, \sigma)$$

Hence, (32) gives

$$-(1-t)\mathcal{W}^2(\mu,\nu) = \int \langle v_t(x), x-y\rangle \, d\bar{\gamma}(x,y) = -(1-t)\int \langle v_t((1-t)x+ty), y-x\rangle \, d\gamma(x,y)$$

$$\tag{33}$$

$$\Rightarrow \quad \mathcal{W}^2(\mu,\nu) = \int \langle v_t((1-t)x+ty), y-x\rangle \, d\gamma(x,y) \tag{34}$$

For $t$ satisfying (31), the fact that $\|v_t\|_{L^2(\mu_t)} = \mathcal{W}(\mu,\nu)$ and $\|y-x\|_{L^2(\gamma)} = \mathcal{W}(\mu,\nu)$ means that (34) gives equality for Cauchy-Schwarz. Thus, $v_t((1-t)x+ty) = y-x$, $\gamma$-almost surely and (28) follows for Lebesgue almost every $t \in [0,1]$.

The final bullet is a direct restatement of the results of Section 8.4 in [1]. $\qquad\square$

# B  Proof of iteration complexity (17)

In this section, we provide a proof of Theorem 1. Consider the following lemma which quantifies the stability of the sub-problems that are used in Algorithm 1.

**Lemma 2.** *Let $\gamma \in \Pi(\mu,\nu)$ be an optimal transport plan between $\mu \in \mathcal{P}_2(\mathbb{R}^d)$ and $\nu \in \mathcal{P}_2(\mathbb{R}^d)$. If $\phi_\mu \in C^1(\mathbb{R}^d)$ is L-smooth*

$$\|\nabla\phi_\mu(x) - \nabla\phi_\mu(y)\| \le L\|x-y\| \tag{35}$$

*then*

$$\left| \int_{\mathbb{R}^d} \langle \nabla\phi_\mu(x), y-x\rangle \, d\gamma(x,y) - \left( \int_{\mathbb{R}^d} \phi_\mu \, d\nu - \int_{\mathbb{R}^d} \phi_\mu \, d\mu \right) \right| \le \frac{L}{2}\mathcal{W}^2(\nu,\mu) \tag{36}$$

*Proof.*  First, it will be shown that

$$\left| \int_{\mathbb{R}^d} \langle \nabla\phi_\mu(x), y-x\rangle \, d\gamma(x,y) - \int_0^1 \langle \nabla\phi_\mu, v_t\rangle_{\mu_t} \, dt \right| \le \frac{L}{2}\mathcal{W}^2(\nu,\mu) \tag{37}$$

for $\mu_t$ and $v_t$ which correspond (26) to the unique-constant speed geodesic given by $\gamma \in \Pi(\mu,\nu)$ (9). Notice that, since $\nabla\phi_\mu$ has at most linear growth, therefore both terms in the left-hand side of (37) are finite. Moreover, by (28), one has

$$\int_0^1 \langle \nabla\phi_\mu, v_t\rangle_{\mu_t} \, dt = \int_0^1 \int_{\mathbb{R}^d \times \mathbb{R}^d} \langle \nabla\phi_\mu((1-t)x+ty), y-x\rangle \, d\gamma(x,y) \, dt \tag{38}$$

Thus, Cauchy-Schwarz and (35) give

$$\left| \int_0^1 \langle \phi_\mu, v_t\rangle_{\mu_t} \, dt - \int_{\mathbb{R}^d \times \mathbb{R}^d} \langle \nabla\phi_\mu(x), y-x\rangle \, d\gamma(x,y) \right| =$$

$$\left| \int_0^1 \int_{\mathbb{R}^d \times \mathbb{R}^d} \langle \nabla\phi_\mu((1-t)x+ty) - \nabla\phi_\mu(x), y-x\rangle \, d\gamma(x,y) \, dt \right| \le$$

$$\int_0^1 \int_{\mathbb{R}^d} tL\|x-y\|^2 \, d\gamma(x,y) \, dt = \frac{L}{2}\mathcal{W}^2(\nu,\mu)$$

To obtain (36), it only remains to show that

$$\int_0^1 \langle \nabla\phi_\mu, v_t\rangle_{\mu_t} \, dt = \int \phi_\mu \, d\nu - \int \phi_\mu \, d\mu \tag{39}$$

Moreover, since $v_t$ satisfies (27), Lemma 8.1.2 in [1] gives

$$\int_0^1 \langle \nabla\psi, v_t\rangle_{\mu_t} \, dt = \int \psi \, d\nu - \int \psi \, d\mu \tag{40}$$

for every $\psi \in C_c^1(\mathbb{R}^d)$– where $C_c^1(\mathbb{R}^d)$ denotes the space of continuously differentiable functions on $\mathbb{R}^d$ with compact support. Hence, (39) will be obtained from (40) by the following approximation argument.

Define the functions:

$$\beta_-(x) := \left(\sqrt{\|x\|^2 + 1} - \sqrt{2}\right)^{-1} \quad \text{and} \quad \beta_+(x) := \left(\sqrt{5} - \sqrt{\|x\|^2 + 1}\right)^{-1}$$

and

$$\eta(x) := \begin{cases} 1 & \text{if } \|x\| \leq 1 \\ \frac{e^{\beta_-(x)}}{e^{\beta_-(x)} + e^{\beta_+(x)}} & \text{if } 1 < \|x\| < 2 \\ 0 & \text{if } \|x\| \geq 2 \end{cases}$$

It is easy to verify that $\eta \in C_c^\infty\left(\mathbb{R}^d\right)$ and $\|\nabla \eta(x)\| \leq B$ for all $x \in \mathbb{R}^d$ and some constant $B$. Moreover, $\eta$ provides a sequence of mollified approximations of $\phi_\mu$

$$\psi_k(x) := \phi_\mu(x)\eta_k(x) \quad \text{for} \quad \eta_k(x) := \eta\left(\frac{x}{k}\right)$$

where $\psi_k \in C_c^1(\mathbb{R}^d)$. Clearly, (40) holds for all such $\psi_k$. Thus, if

$$\lim_{k \to \infty} \int \psi_k \, d\nu - \int \psi_k \, d\mu = \int \phi_\mu \, d\nu - \int \phi_\mu \, d\mu \tag{41}$$

and

$$\lim_{k \to \infty} \int_0^1 \langle \nabla \psi_k, v_t \rangle_{\mu_t} \, dt = \int_0^1 \langle \nabla \phi_\mu, v_t \rangle_{\mu_t} \, dt \tag{42}$$

then (39) will follow directly from (40).

The relations (41) and (42) are straight-forward consequences of dominated convergence. Indeed, as $\eta_k \to 1$ and $\nabla \eta_k \to 0$ (pointwise), clearly

$$\psi_k \to \phi_\mu \quad \text{and} \quad \nabla \psi_k \to \nabla \phi_\mu \tag{43}$$

Quadratic growth of $\phi_\mu$ yields $\phi_\mu \in L^2(\mu) \cap L^2(\nu)$ and combined with

$$|\psi_k(x)| \leq |\phi_\mu(x)| \quad \forall x \in \mathbb{R}^d$$

(41) clearly holds via dominated convergence. One also has

$$\|\nabla \psi_k(x)\| \leq \|\nabla \phi_\mu(x)\| + \frac{B|\phi_\mu(x)|}{k}\mathbf{1}_{\{\|x\| < 2k\}} \tag{44}$$

Using the quadratic growth of $\phi_\mu$, linear growth of $\|\nabla \phi_\mu\|$, and the bound $\|x\|\,\mathbf{1}_{\{\|x\| < 2k\}}/k \leq 2$, (44) yields

$$\|\nabla \psi_k(x)\| \leq \|\nabla \phi_\mu(x)\| + C\,\|x\|\,\mathbf{1}_{\{\|x\| < 2k\}} + D \leq E\,\|x\| + F \tag{45}$$

for some constants $C, D, E \in \mathbb{R}_+$. Recalling (38), (45) provides

$$\int_{\mathbb{R}^d \times \mathbb{R}^d} |\langle \nabla \psi_k((1-t)x + ty), y - x \rangle| \, d\gamma(x,y) \leq \int_{\mathbb{R}^d \times \mathbb{R}^d} \|\nabla \psi_k((1-t)x + ty)\|\,\|y - x\| \, d\gamma(x,y)$$

$$\leq \int_{\mathbb{R}^d \times \mathbb{R}^d} (E\,\|(1-t)x + ty\| + F)\,\|y - x\| \, d\gamma(x,y)$$

$$\leq H \tag{46}$$

for some $H \in \mathbb{R}_+$; where the last inequality is a result of Cauchy-Schwarz. The combination of pointwise convergence (43) and (46) then immediately yield (42) by dominated convergence and (38). $\qquad \square$

We also require the following elementary results regarding the convergence of certain polynomial sequences.

**Lemma 3.** *Let $r_i \in \mathbb{R}_+$ be a sequence of non-negative numbers satisfying*

$$r_{i+1} \leq r_i - \kappa r_i^p \tag{47}$$

*for some constants $\kappa > 0$ and $p \geq 0$. Then,*

$$r_n \leq \begin{cases} e^{-\kappa n / r_0^{1-p}} r_0 & \text{if } p \leq 1 \\ \left(\kappa n + r_0^{1-p}\right)^{-1/(p-1)} & \text{if } p > 1 \end{cases} \tag{48}$$

*Proof.* If $p \leq 1$, then (47) combined with the fact that $r_i$ is a non-increasing sequence implies

$$r_i \leq \left(1 - \frac{\kappa}{r_0^{1-p}}\right) r_{i-1}$$

Iterating this inequality from 1 to $n$ yields the first part of (48). Next, let $p > 1$ and notice that, by taking the reciprocals of both sides of (47) and rearranging, one obtains

$$\frac{\kappa r_{i-1}^{p-2}}{1 - \kappa r_{i-1}^{p-1}} \leq r_i^{-1} - r_{i-1}^{-1}$$

Summing this inequality over $i$ (from 1 to $n$),

$$\kappa n r_k^{p-2} \leq \sum_{i=1}^{n} \frac{\kappa r_{i-1}^{p-2}}{1 - \kappa r_{i-1}^{p-1}} \leq r_k^{-1} - r_0^{-1}$$

where the first inequality is a result of $r_i$ being non-increasing. Algebraic manipulation then provides

$$r_n \leq \left(\kappa n + r_0^{1-p}\right)^{-1/(p-1)}$$

$\square$

*Proof of Theorem 1.* Recall the parameters specified in Assumptions 1 and 3 and let $\epsilon$ be the desired tolerance with which (16) should hold. Let Algorithm 1 be run with the following parameters:

$$\beta_1 = \min\left(\Delta_1, \Delta_2\right), \quad \beta_2 = \alpha(4L)^{-1}, \quad \beta_3 = (1 - \alpha/2)^{1/\alpha} T^{-1/\alpha} \tag{49}$$

and

$$r = \tau \epsilon^\theta / 2, \quad \hat{\epsilon} = (2\alpha^*)^{-1} r, \quad \bar{\epsilon} = \alpha r / 2, \quad \tilde{\epsilon} = (4\alpha^*)^{-1} r, \quad k = \lceil M \rceil \tag{50}$$

where $\alpha^* = (1 + \alpha)/\alpha$ is the dual exponent of $1 + \alpha$ and $M$ is defined in (65). It will be shown that the last iterate, $\mu_l$, computed by Algorithm 1 satisfies (16).

First, we bound the decrease in $J$ at each step of Algorithm 1. Let $\delta_i$ be the $i$th value of $\delta$ that is computed by Algorithm 1 and let $s_i$ denote the $i$th value of $s$. One has the relation

$$\delta_i = \min\left(\beta_1, \beta_2 s_i, \beta_3 s_i^{\alpha^* - 1}\right) \tag{51}$$

and, since $\delta_i \leq \Delta_2$ for all $i$, $\mu_0 \in S$ implies $\mu_i \in S$ for all $i$. Via the smoothness of $J$ on $S$ and $\delta_i \leq \Delta_1$, it follows that

$$J(\mu_i) \leq J(\mu_{i-1}) + \int_{\mathbb{R}^d \times \mathbb{R}^d} \langle F(\mu_{i-1}; x), y - x \rangle \, d\gamma(x, y) + \frac{T}{1 + \alpha} \delta_i^{1+\alpha}$$

for any optimal transport plan $\gamma \in \Pi(\mu_i, \mu_{i-1})$ between $\mu_i$ and $\mu_{i-1}$. Recognizing (12),

$$\int_{\mathbb{R}^d \times \mathbb{R}^d} \left\langle F(\mu_{i-1}; x) - \nabla\widehat{\phi}_{\mu_{i-1}}(x), y - x \right\rangle \, d\gamma(x, y) \leq \left\| F(\mu_{i-1}; x) - \nabla\widehat{\phi}_{\mu_{i-1}} \right\|_{L^2(\mu_{i-1})} W(\mu_i, \mu_{i-1}) \leq \delta_i \hat{\epsilon}$$

and therefore

$$J(\mu_i) \leq J(\mu_{i-1}) + \int_{\mathbb{R}^d \times \mathbb{R}^d} \nabla\widehat{\phi}_{\mu_{i-1}}(x)^T (y - x) \, d\gamma(x, y) + \frac{T}{1 + \alpha} \delta_i^{1+\alpha} + \delta_i \hat{\epsilon}$$

Via Lemma 2,

$$J(\mu_i) \leq J(\mu_{i-1}) + \int \widehat{\phi}_{\mu_{i-1}} \, d\mu_i - \int \widehat{\phi}_{\mu_{i-1}} \, d\mu_{i-1} + \frac{T}{1 + \alpha} \delta_i^{1+\alpha} + \frac{L}{2} \delta_i^2 + \delta_i \hat{\epsilon} \tag{52}$$

Now, since $\widehat{\phi}_{\mu_{i-1}}$ is $L$-smooth, it is a Kantorovich potential [1, Section 6.1] for $\mu_{i-1}$– under the cost function $L \|x - y\|^2 / 2$. Thus, there exists a geodesic $\nu_t$ (Proposition 2) such that: $\nu_0 = \mu_{i-1}$ and the transport plan $\gamma_t \in \Pi(\mu_{i-1}, \nu_t)$ between $\mu_{i-1}$ and $\nu_t$ satisfies [1, Section 8.3]

$$\int_{\mathbb{R}^d \times \mathbb{R}^d} \left\langle \nabla\phi_{\mu_{i-1}}(x), y - x \right\rangle \, d\gamma_t(x, y) = -\frac{t}{L} \left\| \nabla\widehat{\phi}_{\mu_{i-1}} \right\|_{L^2(\mu_{i-1})}^2 \quad \text{and} \quad \mathcal{W}(\nu_t, \mu_{i-1}) = \frac{t}{L} \left\| \nabla\widehat{\phi}_{\mu_{i-1}} \right\|_{L^2(\mu_{i-1})}$$

for $0 \leq t \leq 1$. For the sake of notation, define $g_{i-1} := \left\|\nabla\widehat{\phi}_{\mu_{i-1}}\right\|_{L^2(\mu_{i-1})}$ and set $t = L\delta_i/g_{i-1}$.
Clearly, $t \leq 1$ since $\delta_i \leq \beta_2 s_i \leq \beta_2 g_{i-1}$.

By construction, $\mu_i$ also satisfies

$$\int \widehat{\phi}_{\mu_{i-1}}\, d\mu_i - \int \widehat{\phi}_{\mu_{i-1}}\, d\mu_{i-1} \leq \int \widehat{\phi}_{\mu_{i-1}}\, d\nu_t - \int \widehat{\phi}_{\mu_{i-1}}\, d\mu_{i-1} + \zeta_i$$

for $\zeta_i = \delta_i\widetilde{\epsilon}$. Hence, with another application of Lemma 2, one obtains

$$\int \widehat{\phi}_{\mu_{i-1}}\, d\mu_i - \int \widehat{\phi}_{\mu_{i-1}}\, d\mu_{i-1} \leq \int_0^t \left\langle\nabla\widehat{\phi}_{\mu_{i-1}}, v_s\right\rangle_{\nu_s}\, ds + \zeta_i$$

$$\leq \int_{\mathbb{R}^d\times\mathbb{R}^d} \left\langle\nabla\widehat{\phi}_{\mu_{i-1}}(x), y - x\right\rangle\, d\gamma(x,y) + \frac{L}{2}\mathcal{W}(\nu_t, \mu_{i-1})^2 + \zeta_i$$

$$= -\frac{t}{L}\left(1 - \frac{t}{2}\right) g_{i-1}^2 + \zeta_i \tag{53}$$

Combining (53) with (52) and recalling $\delta_i = tg_{i-1}/L$ gives

$$J(\mu_i) \leq J(\mu_{i-1}) - \frac{t}{L}\left(C - t - \frac{D}{1+\alpha}t^\alpha\right) g_{i-1} + \zeta_i \tag{54}$$

for the values

$$C := 1 - \hat{\epsilon} \quad\text{and}\quad D := \frac{T}{L^\alpha g_{i-1}^{1-\alpha}}$$

Rewriting (54) using the residual term

$$r(\nu) := J(\nu) - \inf_{\mu\in S} J(\mu) \tag{55}$$

one obtains

$$r(\mu_i) \leq r(\mu_{i-1}) - \frac{t}{L}\left(C - t - \frac{D}{1+\alpha}t^\alpha\right) g_{i-1} + \zeta_i \tag{56}$$

This relation will now be used to show that Algorithm 1 makes sufficient progress on $J$, prior to the termination of it's loop.

Let $l$ be the index of the last iterate $\mu_i$ which is computed by Algorithm 1. First, observe that if $s_{l+1} \leq r$, then early termination of the loop in Algorithm 1 has occurred. Using (14) and the definitions (50), it follows that

$$\tau\left(r(\mu_l)\right)^\theta \leq \|F(\mu_l)\|_{L^2(\mu_l)} \leq g_l + \hat{\epsilon}$$

$$\leq r + \bar{\epsilon} + \hat{\epsilon} \leq \tau\epsilon^\theta \tag{57}$$

and, hence, sufficient progress on $J$ has been made– $\mu_l$ satisfies (16). Thus, we need only analyze the case where early termination in Algorithm 1 does not occur and $l = k$ (50).

If $l = k$, then $s_i > r$ for all $i \leq k$ and, by extension, $g_{i-1} > r$ for all $i \leq k$ since $s_i$ is a lower bound for $g_{i-1}$. In this case, the definitions of $\hat{\epsilon}$ and $r$ (50) imply $C \geq 1 - \alpha/\left(2(1+\alpha)\right)$ and the choices for $\beta_2$ and $\beta_3$ (49) provide

$$t \leq \min\left(\frac{\alpha}{2(1+\alpha)}, \frac{(1-\alpha/2)^{1/\alpha}}{D^{1/\alpha}}\right)$$

This gives

$$C - t - \frac{D}{(1+\alpha)}t^\alpha \geq (2\alpha^*)^{-1}$$

from which substitution into (56) yields

$$r(\mu_i) \leq r(\mu_{i-1}) - \frac{t}{2L\alpha^*}g_{i-1} + \zeta_i$$

$$\leq r(\mu_{i-1}) - \frac{\delta_i}{2\alpha^*}g_{i-1} + \zeta_i$$

$$\leq r(\mu_{i-1}) - \frac{\delta_i}{4\alpha^*}g_{i-1} \tag{58}$$

where the last inequality is a result of the definition of $\widetilde{\epsilon}$ (50), $\zeta_i$, and $g_{i-1} > r$. As $\delta_i$ is the minimum of three different terms (51), (58) will be used to analyze the amount of progress, that is made on the objective $J$, corresponding to each of these three terms. Note, the following identities that will be used in the analysis of each term:

$$\left(1 - \frac{\alpha}{2}\right) g_{i-1} \leq g_{i-1} - \frac{\alpha r}{2} \leq g_{i-1} - \bar{\epsilon} \leq s_i \tag{59}$$

and

$$-g_{i-1}^p \leq -\left(\|F(\mu_{i-1})\|_{L^2(\mu_{i-1})} - \hat{\epsilon}\right)^p$$
$$\leq -\left(1 - \frac{\alpha}{2+\alpha}\right)^p \|F(\mu_{i-1})\|_{L^2(\mu_{i-1})}^p \leq -\frac{1}{2e} \|F(\mu_{i-1})\|_{L^2(\mu_{i-1})}^p \tag{60}$$

for all $1 \leq p \leq \alpha^*$. The relation (59) simply observes that $s_i$ is a multiplicative approximation to $g_{i-1}$ in Algorithm 1, while (60) is a consequence of $r - \hat{\epsilon} \leq \|F(\mu_{i-1})\|_{L^2(\mu_{i-1})}$.

First, consider the case where $\delta_i = \beta_1$. Substitution into (58), coupled with (60), provides

$$r(\mu_i) \leq r(\mu_{i-1}) - \frac{\beta_1}{8e\alpha^*} \|F(\mu_{i-1})\|_{L^2(\mu_{i-1})} \tag{61}$$

Applying (14) to (61) and defining $r_i := r(\mu_i)$ (for the sake of notation) yields

$$r_i \leq r_{i-1} - \kappa^{(1)} r_{i-1}^\theta \qquad \text{for} \qquad \kappa^{(1)} := \omega\beta_1 \tag{62}$$

for the constant $\omega = (8e\alpha^*)^{-1}\tau$. In the cases (51) corresponding to $\beta_2$ and $\beta_3$, similar applications of the previous identities (along with (59)) give

$$r_i \leq r_{i-1} - \kappa^{(2)} r_{i-1}^{2\theta} \qquad \text{for} \qquad \kappa^{(2)} := \omega\tau(1-\alpha/2)\beta_2 \tag{63}$$

$$r_i \leq r_{i-1} - \kappa^{(3)} r_{i-1}^{\alpha^*\theta} \qquad \text{for} \qquad \kappa^{(3)} := \omega\left(\tau(1-\alpha/2)\right)^{1/\alpha}\beta_3 \tag{64}$$

Now, for the sake of notation, define the function

$$z(u,v) := u^{-1}\epsilon^{-(1-v)_-}\left(r_0 \log^{1/(1-v)}(r_0/\epsilon)\right)^{(1-v)_+}$$

where $(\cdot)_+$ and $(\cdot)_-$ denote the positive and negative parts. Using Lemma 3, it follows that, if (62) occurs for more than $\omega^{-1}z(\beta_1, \theta)$ iterations of Algorithm 1, then $r_k \leq \epsilon$, where $k$ is the index of the last loop iteration in Algorithm 1. Similar deductions for (63) and (64) lead to the conclusion that, if

$$k \geq \omega^{-1}\left(z(\beta_1,\theta) + z(\tau(1-\alpha/2)\beta_2, 2\theta) + z((\tau(1-\alpha/2))^{1/\alpha}\beta_3, \alpha^*\theta)\right) := M \tag{65}$$

then either (62), (63), or (64) has occurred sufficiently many times during the execution of Algorithm 1 to guarantee $r_k \leq \epsilon$. As $k$ has been chosen exactly so that $k = \lceil M \rceil$ (50), one obtains that $\mu_k$ satisfies (16). The desired complexity bound (17) on $M$ now follows by plugging in for $\beta_1, \beta_2$, and $\beta_3$ in (65) and then, taking asymptotic estimates as $\epsilon \to 0$; the term $z((\tau(1-\alpha/2))^{1/\alpha}\beta_3, \alpha^*\theta)$ clearly dominates. $\qquad\square$

## C  Computational solution of the Frank-Wolfe problem (15)

This section provides a concrete, computational procedure and complexity guarantee for the subroutine (15) in Algorithm 1 which, for $\mu \in \mathcal{P}_2(\mathbb{R}^d)$, requires solution of the problem

$$\inf_{\nu \in \mathcal{P}_2(\mathbb{R}^d), \mathcal{W}(\nu,\mu)\leq\delta} \int f \, d\nu \tag{66}$$

The starting point to solve this problem is duality. The dual of (66) is

$$D(f) := \sup_{\lambda \in \mathbb{R}_+}\left[\int_{\mathbb{R}^d} f_\lambda(x)\, d\mu(x) - (\delta^2\lambda)/2\right], \qquad \text{where } f_\lambda(x) := \inf_{y \in \mathbb{R}^d} f(y) + \frac{\lambda}{2}\|x-y\|^2 \tag{67}$$

where $f_\lambda$ is the Moreau-Yosida envelope [62] for $f$. The problem (67) permits practical computation since it requires only finite dimensional optimization procedures to calculate $f_\lambda$ and perform ascent in $\lambda$. Moreover, strong duality between (66) and (67) holds under quite general circumstances [6] and, particularly for any of the circumstances in this work where $f$ is assumed to be smooth.

Previous work [6, 54] has noted that, in general, solution of (67) might still be computationally infeasible for smooth $f$; for arbitrary $\lambda$, $f_\lambda$ could obscure a computationally difficult problem with many local minima. However, for large enough $\lambda$, $f_\lambda$ is quite computable since it's defining minimization problem becomes convex. So long as all relevant $\lambda$ in (67) are large enough, this means that (67) will be efficiently computable. This is equivalent to ensuring that the trust-region size $\delta$ in (66) is not too large.

With the following results, we establish a bound on $\delta$ which is simultaneously small enough to achieve computational tractability for (67), but large enough to permit the iterative complexities of Theorem 1. Algorithms 2 and 3 are also provided to leverage these results and yield a computational procedure with concrete complexity for solving (66). For simplicity, rewrite (67) as

$$D(f) = \sup_{\lambda \in \mathbb{R}} g(\lambda) - \left(\delta^2 \lambda\right)/2, \qquad \text{where } g(\lambda) := \int_{\mathbb{R}^d} f_\lambda \, d\mu \tag{68}$$

Recall that a function $\phi : \mathbb{R}^d \to \bar{\mathbb{R}}$ is called *semiconvex* if

$$x \longrightarrow \phi(x) + \frac{\lambda}{2} \|x - x_0\|^2 \tag{69}$$

is convex for some $\lambda \geq 0$ and some $x_0 \in \mathbb{R}^d$. Further, a continuously differentiable function $\phi$ is called $L$-smooth if it has $L$-Lipschitz gradients:

$$\|\nabla \phi(y) - \nabla \phi(x)\| \leq L \|y - x\| \tag{70}$$

**Lemma 4.** *If $f$ is differentiable and $\rho_*$-semiconvex (69), the function $g$ (68) is differentiable on $(\rho_*, \infty)$ and*

$$g'(\lambda) = \frac{1}{2} \int_{\mathbb{R}^d} \left\|y^*_{\lambda,x} - x\right\|^2 \, d\mu(x), \quad y^*_{\lambda,x} := \arg\min_{y \in \mathbb{R}^d} f(y) + \frac{\lambda}{2} \|y - x\|^2 \tag{71}$$

*where the unique minimizer $y^*_{\lambda,x}$ satisfies*

$$\frac{1}{2} \left\|y^*_{\lambda,x} - x\right\|^2 \leq \frac{2}{(\lambda - \rho_*)^2} \|\nabla f(x)\|^2 \tag{72}$$

*Additionally, for any $\rho_* < \lambda_1 \leq \lambda_2$ one has*

$$\left(1 - 2\sqrt{\frac{\lambda_2 - \lambda_1}{\lambda_2 - \rho_*}}\right) g'(\lambda_1) \leq g'(\lambda_2) \tag{73}$$

*That is, for any $t^* > \rho_*$, $g'$ is $1/2$-Holder continuous on $[t^*, \infty)$ with a constant depending only on $t^*$ and $\rho_*$.*

*Proof.* Define the functions

$$a_\lambda(y; x) := f(y) + \frac{\lambda}{2} \|y - x\|^2 \qquad \text{and} \qquad z_x(\lambda) := \inf_{y \in \mathbb{R}} a_\lambda(y; x)$$

Since $f$ is $\rho_*$-semiconvex (69), $a_\lambda(y; x)$ is $\lambda - \rho_*$ strongly convex in $y$ for $\lambda > \rho_*$. Therefore, the minimizer $y_{\lambda,x}$ is unique. Further, semiconvexity and differentiability of $f$ provide the lower bound

$$a_\lambda(y; x) \geq f(x) + l_\lambda(y; x) \qquad \text{where} \qquad l_\lambda(y; x) := \nabla f(x)^T (y - x) + \frac{\lambda - \rho_*}{2} \|y - x\|^2$$

Noticing $l_\lambda(y; x) > 0$ for any $y \in \mathbb{R}^d$ such that $\|y - x\| > (2 \|\nabla f(x)\|) / (\lambda - \rho_*)$, one obtains (72).

For open subsets $O \subset (\rho_*, \infty)$ whose closure does not contain $\rho_*$, (72) implies that the radius of the ball containing $y^*_{\lambda,x}$ is uniformly bounded for all $\lambda \in O$. Danskin's theorem [30] can, therefore, be

applied to the function $z_x(\lambda) := f_\lambda(x)$ (67) to conclude that $z_x(\lambda)$ is differentiable on $(\rho_*, \infty)$ with derivative

$$z_x'(\lambda) = \frac{1}{2} \left\| y_{\lambda,x}^* - x \right\|^2$$

Observing that $g(\lambda) = \mathbb{E}_{x \sim \mu} [z_x(\lambda)]$, the conclusion (71) then follows from (72) and dominated convergence.

Finally, let $\rho_* < \lambda_1 \leq \lambda_2$. Since $z_x(\lambda)$ is concave in $\lambda$

$$|z_x'(\lambda_1) - z_x'(\lambda_2)| = z_x'(\lambda_1) - z_x'(\lambda_2)$$

and it is enough to show a one-sided bound on the quantity $z_x'(\lambda_1) - z_x'(\lambda_2)$. To this end, observe

$$z_x'(\lambda_1) - z_x'(\lambda_2) \leq \left\| y_{\lambda_1,x}^* - x \right\| \left\| y_{\lambda_2,x}^* - y_{\lambda_1,x}^* \right\| \tag{74}$$

Hence, (73) can be provided by producing a bound on $\left\| y_{\lambda_2,x}^* - y_{\lambda_1,x}^* \right\|$. Strong convexity of $a_\lambda(y; x)$ in $y$ yields the identity

$$a_{\lambda_2}(y_{\lambda_2,x}^*; x) + \frac{\lambda_2 - \rho_*}{2} \left\| y_{\lambda_2,x}^* - y_{\lambda_1,x}^* \right\|^2 \leq a_{\lambda_2}(y_{\lambda_1,x}^*; x) = a_{\lambda_1}(y_{\lambda_1,x}^*; x) + \frac{\lambda_2 - \lambda_1}{2} \left\| y_{\lambda_1,x}^* - x \right\|^2$$

which, when combined with the fact that $a_{\lambda_1}(y_{\lambda_1,x}^*; x) \leq a_{\lambda_2}(y_{\lambda_1,x}^*; x)$ ($z_x(\lambda)$ is non-decreasing in $\lambda$), gives

$$\left\| y_{\lambda_2,x}^* - y_{\lambda_1,x}^* \right\| \leq \sqrt{\frac{\lambda_2 - \lambda_1}{\lambda_2 - \rho_*}} \left\| y_{\lambda_1,x}^* - x \right\| \tag{75}$$

Applying (75) to (74) and rearranging produces

$$\left( 1 - 2\sqrt{\frac{\lambda_2 - \lambda_1}{\lambda_2 - \rho_*}} \right) z_x'(\lambda_1) \leq z_x'(\lambda_2) \tag{76}$$

Taking the expectation with respect to $x$ on both sides of (76) yields (73). $\qquad\square$

**Lemma 5.** *If $f$ is $L$-smooth (70) and $L < \lambda$ then any optimizer*

$$f(y^*) + \frac{\lambda}{2} \|y^* - x\|^2 = \inf_{y \in \mathbb{R}^d} f(y) + \frac{\lambda}{2} \|y - x\|^2$$

*satisfies* $\|y^* - x\| \geq \frac{\|\nabla f(x)\|}{2\lambda}$.

*Proof.* From $L$-smoothness and the fact $\lambda > L$, the function

$$v(y) := f(y) + \frac{\lambda}{2} \|y - x\|^2$$

is $(\lambda - L)$-strongly convex. To show $\|\nabla f(x)\| / (2\lambda) \leq \|y^* - x\|$ notice

$$\nabla f(y^*) + \lambda(y^* - x) = 0 \tag{77}$$

by first-order optimality conditions for $y^*$. Combining (77) with the $L$-smoothness of $f$, one obtains

$$\|\nabla f(x) - \nabla f(y^*)\|^2 \leq L^2 \|x - y^*\|^2$$
$$\Rightarrow \|\nabla f(x)\|^2 + (\lambda^2 - L^2) \|x - y^*\|^2 \leq 2\lambda \nabla f(x)^T (x - y^*) \leq 2\lambda \|\nabla f(x)\| \|x - y^*\| \tag{78}$$

Using the fact that $\lambda > L$, the desired result then follows directly from (78). $\qquad\square$

The properties provided by Lemma 4 and Lemma 5 now enable establishment of a relationship between trust region size $\delta$ (66) and the decision variables in (67).

**Proposition 3.** *If $f$ is differentiable and $\rho_*$-semiconvex then, for any $\epsilon > 0$, there exists a $\lambda_\epsilon \leq \rho_* + \|\nabla f(x)\|_{L^2(\mu)}^2 / (2\epsilon)$ such that*

$$\left( \sup_{\lambda \in \mathbb{R}} g(\lambda) - \left( \delta^2 \lambda \right) / 2 \right) - \left( g(\lambda_\epsilon) - \left( \delta^2 \lambda_\epsilon \right) / 2 \right) \leq \epsilon \tag{79}$$

*Further, if $f$ is $L$-smooth (70) and $\delta = \|\nabla f(x)\|_{L^2(\mu)} / C$ for $C \geq 2L$, then $\lambda_\epsilon$ can be chosen in the interval $[l, u] \subseteq \mathbb{R}$ for*

$$l = \rho_* \quad and \quad u = \min (\beta, \rho_* + C) \tag{80}$$

*where $\beta = \rho_* + \|\nabla f(x)\|_{L^2(\mu)}^2 / (2\epsilon)$*

*Proof.* For any $\hat\lambda \geq \rho_*$, $\rho_*$-semiconvexity of $f$ and the definition of $g$ (68) provide the lower and upper bounds

$$g(\hat\lambda) = \int_{\mathbb{R}^d} f_\lambda \, d\mu \geq \int_{\mathbb{R}^d} f \, d\mu - \frac{\|\nabla f(x)\|^2_{L^2(\mu)}}{2(\hat\lambda - \rho_*)} \qquad \text{and} \qquad g(\lambda) \leq \int f \, d\mu, \ \ \forall \lambda \in \mathbb{R}$$

These allow one to obtain the identity

$$g(\hat\lambda) - \left(\delta^2\hat\lambda\right)/2 \geq \left(g(\lambda) - \left(\delta^2\lambda\right)/2\right) - \frac{\|\nabla f(x)\|^2_{L^2(\mu)}}{2(\hat\lambda - \rho_*)} + \frac{\delta^2\left(\lambda - \hat\lambda\right)}{2} \tag{81}$$

for any $\hat\lambda \geq \rho_* \geq 0$. Via (81), Proposition 3 can be easily established; indeed let us first show (79).

Define $\lambda_n \in \mathbb{R}$ to be an optimizing sequence for (68)

$$\lim_{n\to\infty} g(\lambda_n) - \left(\delta^2\lambda_n\right)/2 = D(f)$$

and set $\beta := \rho_* + \|\nabla f(x)\|^2_{L^2(\mu)}/(2\epsilon)$. Since $g$ is upper-semicontinuous, it is sufficient to show that there exists a $\lambda_\epsilon \leq \beta$ satisfying (79) if $\beta < \liminf_{n\to\infty} \lambda_n$. Since $\beta < \liminf_{n\to\infty} \lambda_n$, one can assume without loss of generality that $\beta < \lambda_n$ for all $n \in \mathbb{N}$. Substituting $\hat\lambda = \beta$ and $\lambda = \lambda_n$ in (81) simplifying provides

$$g(\beta) - \left(\delta^2\beta\right)/2 \geq g(\lambda_n) - \left(\delta^2\lambda_n\right)/2 - \epsilon \tag{82}$$

Taking the limit in (82) and setting $\lambda_\epsilon = \beta$ gives the desired result (79).

To show the second half of Proposition 3, observe that the previous result implies one can assume $\liminf_{n\to\infty} \lambda_n \leq \beta$ for an optimizing sequence $\lambda_n$– otherwise, $\beta$ is $\epsilon$-optimal. The immediate consequence of this assumption is that an optimizer $\lambda^*$ of (68) exists. Indeed, $L$-smoothness of $f$ provides $g(\lambda) = -\infty$ for any $\lambda < -L$ and, combined with $\liminf_{n\to\infty} \lambda_n \leq \beta$, the optimizing sequence $\lambda_n$ can be assumed to be bounded. Via Bolzano-Weierstrass, the sequence is therefore convergent to some $\lambda^* \leq \beta$ and upper-semicontinuity of $g$ along with lower-semicontinuity of $\psi^*$ then imply that $\lambda^*$ is an optimizer of (68).

The main consequence of the existence of $\lambda^*$ is that, in combination with (81), one has the upper bound

$$\frac{\delta^2(\lambda^* - \lambda)}{2} - \frac{1}{2(\lambda - \rho_*)_+} \|\nabla f(x)\|^2_{L_2(\mu)} + g(\lambda^*) - \left(\delta^2\lambda^*\right)/2 \leq g(\lambda) - \left(\delta^2\lambda\right)/2$$

$$\Rightarrow \quad \frac{\delta^2\left(\lambda^* - \lambda\right)}{2} \leq \frac{\|\nabla f(x)\|^2_{L_2(\mu)}}{2(\lambda - \rho_*)_+}$$

$$\Rightarrow \quad \delta^2(\lambda - \rho_*)_+ \left(\lambda^* - \lambda\right) \leq \|\nabla f(x)\|^2_{L_2(\mu)} \tag{83}$$

if $\lambda \leq \lambda^*$. Taking $\lambda = (\lambda^* + \rho_*)/2$ above will lead to the desired conclusion of Proposition 3– so long as $\rho_* \leq \lambda^*$. To show that $C \geq 4L$ implies $\rho_* \leq \lambda^*$, observe that Lemma 5, in combination with Lemma 4, implies

$$\frac{1}{8\lambda^2} \|\nabla f(x)\|^2_{L^2(\mu)} \leq \partial_+ g(\lambda), \quad \lambda \geq L \tag{84}$$

where $\partial_+(\cdot)$ denotes the derivative of $g$ from the right. Under $C \geq 4L$, (84) produces the relation

$$\frac{\delta^2}{2} \leq \frac{1}{8L^2} \|\nabla f(x)\|^2_{L^2(\mu)} \leq \partial_+ g(L) \leq \partial_+ g(\rho_*) \tag{85}$$

since $\rho_* \leq L$. As $g$ is concave, (85) immediately gives $g(\rho_*) - \left(\delta^2\rho_*\right)/2 \geq g(\lambda) - \left(\delta^2\lambda\right)/2$ for all $\lambda < \rho_*$. Hence, $\lambda^*$ can be chosen so that $\rho_* \leq \lambda^*$. Finally, using the fact that $\rho_* \leq \lambda^*$ and substituting $\lambda = (\lambda^* + \rho_*)/2$ into (83), one obtains

$$\lambda^* \leq \rho_* + \left(\frac{4\|\nabla f(x)\|^2_{L^2(\mu)}}{\delta^2}\right)^{1/2} \leq \rho_* + C \tag{86}$$

After combining (86) with the bounds $\rho_* \leq \lambda^*$ and $\lambda^* \leq \beta$, the final conclusion of Proposition 3 follows. $\qquad\square$

With the bounds of Proposition 3 in hand, a suitable gradient oracle for $g$ (68) can be provided.

**Definition 5** (Gradient oracle with high probability). A function $\theta_g : \mathbb{R} \to \mathbb{R}$ is called a $(\epsilon, \gamma)$-gradient oracle *with high probability* for $g$ if, when queried with a $\lambda$, it returns an independent random sample $\theta_g(\lambda)$ satisfying

$$\mathbb{P}\left(\left[|\theta_g(\lambda) - g'(\lambda)| \geq \frac{\epsilon}{\max(\lambda - l, 1)}\right]\right) \leq \gamma \tag{87}$$

---

**Algorithm 2** Gradient oracle for $g$ (68)

**Input:** Distribution $\mu$, point $\lambda$, semi-convexity parameter $\rho_*$, smoothness parameter $L$, error tolerance $\epsilon$

Sample $x \sim \mu$

$y_0 \leftarrow x, \kappa \leftarrow \sqrt{(\lambda + L)/(\lambda - \rho_*)}$

$k \leftarrow \max\left(\lceil 4\kappa \log(12\kappa \|\nabla f(x)\| / \epsilon)\rceil, 0\right)$

**for** $1 \leq i \leq k$ **do**

$\quad z_i = y_{i-1} - \frac{1}{\kappa}\left(\nabla f(y_{i-1}) + \lambda(y_{i-1} - x)\right)$

$\quad y_i = z_i + \frac{\kappa - 1}{\kappa + 1}(z_i - z_{i-1})$

**return** $\theta = \frac{1}{2}\|y_k - x\|^2$

---

**Proposition 4.** *For a $\rho_*$-semiconvex function $f : \mathbb{R}^d \to \mathbb{R}$, which is also $L \geq \rho_*$ smooth (70), the mean of*

$$K \geq \frac{64 \|\nabla f(x)\|_{L^4(\mu)}^4}{(\lambda - \rho_*)^2 \min\left((\lambda - \rho_*)^2, 1\right)\gamma\tilde{\epsilon}^2} \tag{88}$$

*independent calls to Algorithm 2 with inputs $\lambda > \rho_*$ and $\epsilon = \tilde{\epsilon}/(2\max(\lambda - \rho_*, 1))$, provides a $(\tilde{\epsilon}, \gamma)$-gradient oracle with high probability (5) on the interval $(\rho_*, \infty)$.*

*Proof.* Consider the sample $x$ which is computed by Algorithm 2. In light of Lemma 4, it is clear that

$$\theta^* := \frac{1}{2}\left\|y_{\lambda,x}^* - x\right\|^2$$

is an unbiased estimate of $g'(\lambda)$. Hence, to prove establish Proposition 4, it will first be shown that the output of Algorithm 2, $\theta$, satisfies

$$|\theta - \theta^*| \leq \epsilon \quad \text{and} \quad \theta \leq \left(\frac{4\|\nabla f(x)\|}{\lambda - \rho_*}\right)^2 \tag{89}$$

when $\lambda \in (\rho_*, \infty)$.

To this end, notice that Algorithm 2 performs Nesterov's accelerated gradient descent [48] on the $\lambda - \rho_*$-strongly convex and $\lambda + L$-smooth function $a_\lambda(y; x)$. Strong convexity yields the identity

$$\frac{\lambda - \rho_*}{2}\left\|y_{\lambda,x}^* - y\right\|^2 \leq a_\lambda(y; x) - a_\lambda(y_{\lambda,x}^*; x) \tag{90}$$

while the convergence guarantees of accelerated gradient descent [48, Theorem 2.2.3] give

$$a_\lambda(y_k; x) - a_\lambda(y_{\lambda,x}^*; x) \leq \left(1 - \kappa^{-1}\right)^k (\lambda + L)\left\|y_{\lambda,x}^* - x\right\|^2 \tag{91}$$

for $\kappa = \sqrt{(\lambda + L)/(\lambda - \rho_*)}$. Combining these relations and setting $C = 2\|\nabla f(x)\|/(\lambda - \rho_*)$

$$\left\|y_{\lambda,x}^* - y_k\right\|^2 \leq \frac{2\left(a_\lambda(y_k; x) - a_\lambda(y_{\lambda,x}^*; x)\right)}{\lambda - \rho_*} \leq 2\left(1 - \kappa^{-1}\right)^k \kappa^2 \left\|y_{\lambda,x}^* - x\right\|^2 \leq 2\left(\frac{\epsilon}{6C}\right)^2 \tag{92}$$

since $k \geq 4\kappa \log\left(6\kappa C/\epsilon\right)$ and $\left\|y_{\lambda,x}^* - x\right\| \leq C$ via (72). Completing the analysis,

$$|\theta - \theta^*| = \frac{1}{2}\left|\|y_k - x\|^2 - \|y_{\lambda,x}^* - x\|^2\right| \leq \frac{1}{2}\left\|y_k - y_{\lambda,x}^*\right\|\left(\|y_k - x\| + \|y_{\lambda,x}^* - x\|\right) \tag{93}$$

$$\leq \frac{3}{2}\left\|y_k - y_{\lambda,x}^*\right\|\left\|y_{\lambda,x}^* - x\right\| \leq \epsilon \tag{94}$$

where triangle inequality provides both (93) and

$$\|y_k - x\| \le 2 \left\| y_{\lambda,x}^* - x \right\| \le 2C \tag{95}$$

Moreover, (94) is the desired left-hand inequality of (89) while (95) contains the desired right-hand inequality.

Establishing Proposition 4 is now a straightforward consequence of Chebyshev's inequality using (89). Indeed, one has

$$\left| \mathbb{E}\left[\theta\right] - g'(\lambda) \right| \le \frac{\tilde{\epsilon}}{2 \max\left(\lambda - \rho_*, 1\right)} \quad \text{and} \quad \theta \le \frac{16}{(\lambda - \rho_*)^2} \left\| \nabla f(x) \right\|^2 \tag{96}$$

when $\epsilon = \tilde{\epsilon} / \left( 2 \max\left(\lambda - \rho_*, 1\right) \right)$. Letting $\bar{\theta}$ be the average of $K$ independent calls to Algorithm 2, Chebyshev's inequality gives

$$\mathbb{P}\left( \left| \bar{\theta} - \mathbb{E}\left[\theta\right] \right| \ge \frac{\tilde{\epsilon}}{2 \max(\lambda - \rho_*, 1)} \right) \le \frac{64 \, \left\| \nabla f(x) \right\|_{L^4(\mu)}^4}{(\lambda - \rho_*)^2 \min\left( (\lambda - \rho_*)^2, 1 \right) \tilde{\epsilon}^2 K} \le \gamma \tag{97}$$

$\square$

The supergradient oracle of Proposition 4 provides a mechanism to perform ascent steps in $\lambda$ to solve (68). Indeed, one can now perform bisection ascent for this problem. For the sake of our Frank-Wolfe procedure, it is of importance that this ascent procedure implicitly maintains a primal-feasible iterate for

$$\inf_{\pi \in \Pi(\mu)} \int f \, d\pi + \psi \left( \int \frac{1}{2} \|y - x\|^2 \, d\pi \right) \tag{98}$$

and makes progress on the primal-dual gap between (98) and (67). For this reason, we title the algorithm a "primal-dual" algorithm.

---

**Algorithm 3** Primal-dual, bisection ascent for (68)

---

**Input:** Supergradient oracle $\theta_g$, error tolerance $\epsilon$, termination width $B$
  $\eta \leftarrow \infty, b \leftarrow l$
  **while** $u - l > \epsilon/B$ **do**
    $\lambda \leftarrow (l + u) / 2$
    $\eta \leftarrow \theta_g(\lambda), \eta \leftarrow \left( \eta - (\psi^*)'(\lambda) \right)$
    **if** $\eta < -\epsilon / \max\left(\lambda - b, 1\right)$ **then** $u \leftarrow \lambda$
    **else** $l \leftarrow \lambda$
  **return** $u$

---

*Remark* 7. The primal iterate that this algorithm maintains can be clarified by recalling that the conditions of Proposition 3 guarantee that $\lambda^* > \rho_*$ for any $\rho_*$-semiconvex $f$ and optimal $\lambda^*$ in (67). Since the function $y \mapsto f(y) + \lambda/2 \|x - y\|^2$ is strictly convex for $\lambda > \rho_*$, the distribution given by

$$(X, m(X)) \sim \pi_{\lambda,\mu}, \qquad X \sim \mu, \quad m_\lambda(x) = \arg\min_{y \in \mathbb{R}^d} f(y) + \frac{\lambda}{2} \|y - x\|^2 \tag{99}$$

provide the unique coupling such that

$$\int_{\mathbb{R}^d \times \mathbb{R}^d} f(y) + \frac{\lambda^*}{2} \|y - x\|^2 \, d\pi_{\lambda,\mu}(x, y) = \int_{\mathbb{R}^d} \min_{y \in \mathbb{R}^d} \left( f(y) + \frac{\lambda^*}{2} \|y - x\|^2 \right) d\mu(x) \tag{100}$$

Through $\lambda$, Algorithm 3 implicitly maintains $\pi_{\lambda,\mu}$ and the criterion used for bisection of an interval in Algorithm 3 is designed to make progress on the primal-dual gap between the current dual iterate $\lambda_i$ and $\pi_{\lambda_i,\mu}$:

$$G(\lambda_i) := \int f \, d\pi_{\lambda_i,\mu} + \infty \mathbf{1}_{(\delta,\infty)} \left( \int \|y - x\|^2 \, d\pi_{\lambda_i,\mu} \right) - \left( g(\lambda_i) - \left( \delta^2 \lambda_i \right) / 2 \right) \tag{101}$$

This stands contrary to other approaches [54] for solving (67) where the computed dual feasible iterate $\lambda_i$ need not provide a primal feasible iterate.

**Theorem 6.** *If $\mu \in \mathcal{P}_2\left(\mathbb{R}^d\right)$, $f$ is $L$-smooth (70), and $\delta \leq \|\nabla f\|_{L^2(\mu)}/(2L)$, there exists a stochastic algorithm which (for any probability $\gamma < 1$) computes a $\lambda^*$ such that $\mathcal{W}\left(\nu_{\lambda^*}, \mu\right) \leq \delta$ and*

$$\int f \, d\nu_{\lambda^*} - \inf_{\mathcal{W}(\nu,\mu) \leq \delta} \int f \, d\nu \leq \epsilon \tag{102}$$

*where $\nu_{\lambda^*}$ is second marginal of $\pi_{\lambda^*,\mu}$ in (99). This algorithm requires at most $\widetilde{O}(L^2 \|\nabla f\|_{L^4(\mu)}^4 /((1-\gamma)\epsilon^2))$ independent samples from $\mu$ and executes $\widetilde{O}(L^{5/2} \|\nabla f\|_{L^4(\mu)}^4 /((1-\gamma)\epsilon^2))$ gradient evaluations of $f$ in expectation.*

**Lemma 7.** *Under the assumptions of Theorem 6, let Algorithm 3 be run with a $(\epsilon, \tau)$-gradient oracle (Definition 5) and $B = 4(g'(l))^2$ on the interval $[l, u]$. Recalling (99), if $1 \leq l - L$ and $g'(u) - \delta^2/2 \leq 0 \leq g'(l) - \delta^2/2$, the output $\lambda^*$ of Algorithm 3 satisfies (recall (101))*

$$G(\lambda^*) \leq (4 + l)\epsilon \tag{103}$$

*with probability at least $1 - \tau\left(\log_2\left(B(u - l)/\epsilon\right) + 1\right)$.*

*Proof.* Let $\lambda_i, u_i, l_i$ and $\eta_i$ denote the $i$th values of $\lambda, u, l$ and $\eta$ which are computed by Algorithm 3– the indexes $l_0, u_0$ denote the initial values of these variables. Let $k$ denote the total number of iterations performed by the loop of Algorithm 3. Since $u_i - l_i = (u_{i-1} - l_{i-1})/2$, it is clear that $k \leq \log_2\left(B(u_0 - l_0)/\epsilon\right) + 1$. Thus, using (87) and the fact that $\lambda_i$ depends only on $\theta_g(\lambda_j)$ for $j < i$, one obtains the union bound

$$\mathbb{P}\left(\bigcup_{i \leq k}\left[|\theta_g(\lambda_i) - g'(\lambda_i)| \geq \frac{\epsilon}{\max(\lambda_i - l_0, 1)}\right]\right) \leq \tau\left(\log_2\left(B(u_0 - l_0)/\epsilon\right) + 1\right) \tag{104}$$

Hence, it need only be shown that (103) holds when

$$|\theta_g(\lambda_i) - g'(\lambda_i)| \leq \frac{\epsilon}{\max\left(\lambda_i - l_0, 1\right)} \quad \forall \, i \leq k \tag{105}$$

For brevity, set $\epsilon_{\lambda_i} = \epsilon/\max(\lambda_i - l_0, 1)$ and recall $\eta_i = \theta_g(\lambda_i) - \delta^2/2$. Define $\eta_i^* := g'(\lambda_i) - \delta^2/2$ to be the true supergradient of (68) which $\eta_i$ approximates. From (105)

$$\eta_i \eta_i^* \leq 0 \quad \Rightarrow \quad \max\left(|\eta_i|, |\eta_i^*|\right) \leq \epsilon_{\lambda_i} \tag{106}$$

Hence, at all iterations prior to the last iteration (iteration $k$), $\eta_i$ and $\eta_i^*$ have the same sign. Since $\lambda \mapsto \left(g(\lambda) - \left(\delta^2 \lambda\right)/2\right)$ is concave, this gives

$$\sup_{\lambda \in [l_i, u_i]} g(\lambda) - \left(\delta^2 \lambda\right)/2 = \sup_{\lambda \in [l_{i-1}, u_{i-1}]} g(\lambda) - \left(\delta^2 \lambda\right)/2 \tag{107}$$

for all $1 < i < k$. Additionally, if $\eta_k \eta_k^* > 0$ then (107) also holds for $i = k$.

From Algorithm 3, it is clear that either $u_k = \lambda_i$ for some $i > 0$ such that $\eta_i < -\epsilon_{\lambda_i}$ or $u_k = u_0$. Similarly, $l_k = \lambda_j$ for some $j > 0$ such that $\eta_j \geq -\epsilon_{\lambda_j}$ or $l_k = l_0$. One can assume, without loss of generality, that $l_k = \lambda_j$ for some $j > 0$ and, in combination with (106) and $g'(u_0) - \delta^2/2 \leq 0 \leq g'(l_0) - \delta^2/2$, this gives

$$-\epsilon_{\lambda_j} \leq g'(l_k) - \delta^2/2 \quad \text{and} \quad g'(u_k) - \delta^2/2 \leq 0 \tag{108}$$

Thus,

$$G(u_k) = \int f \, d\pi_{u_k,\mu} - \left(g(u_k) - \left(\delta^2 u_k\right)/2\right) = u_k\left(\delta^2/2 - g'(u_k)\right)$$

and using (73), one obtains

$$G(u_k) \leq u_k\left(\delta^2/2 - \left(1 - \frac{2}{\sqrt{u_k - \rho_*}}(u_k - l_k)^{1/2}\right)g'(l_k)\right)$$

$$\leq u_k \epsilon_{\lambda_j} + \frac{2u_k(u_k - l_k)^{1/2}}{\sqrt{u_k - \rho_*}}g'(l_k) \tag{109}$$

where the last inequality is a result of (108). To bound the first term on the left side of (109), notice that $l_k \neq l_0$ implies there exists a minimal $t > 0$ such that $l_t \neq l_0$. Clearly,

$$u_k \epsilon_{\lambda_j} = \frac{u_k \epsilon}{\max(\lambda_j - l_0, 1)} \leq \frac{u_k \epsilon}{\max(\lambda_t - l_0, 1)} \leq \epsilon \frac{u_t - l_0}{\max(\lambda_t - l_0, 1)} + l_0 \epsilon \leq (2 + l_0)\epsilon$$

Combining this with the termination condition

$$u_k - l_k \leq \frac{\epsilon}{B} \leq \frac{\epsilon}{4g(l_0)^2} \leq \frac{\epsilon}{4g(l_k)^2}$$

to bound the second term of (109), one obtains

$$T \leq (4 + l_0)\epsilon$$

$\square$

*Proof of Theorem 6.* Let $l = L + 1$ and $u = L + 1 + 4L$ and apply Algorithm 3 to the interval $[l, u]$ with the supergradient oracle given by Proposition 4. Set the error tolerance used by Algorithm 3 to $\epsilon/(4 + L + 1)$ and the termination width to $B := 16 \|\nabla f(x)\|_{L^2(\mu)}^4$. Likewise, the error tolerance used in Proposition 4 should be be $\epsilon/(4 + L + 1)$ and the error probability should be $(1 - \gamma)/(\log_2(B(u - l)(4 + L + 1)/\epsilon) + 1)$.

Under this setting of parameters, Lemma 7 establishes that the output of $\lambda^*$ of Algorithm 3 satisfies

$$G(\lambda^*) \leq \epsilon \tag{110}$$

with probability $\gamma$ so long as

$$g'(u) - \delta^2/2 \leq 0 \leq g'(l) - \delta^2/2 \tag{111}$$

To see that (111) is fulfilled for the chosen $l$ and $u$, notice that the conditions of Theorem 6 guarantee that Lemma 4 holds. Hence, (85) gives

$$g'(l) - \delta^2/2 \geq 0$$

Similarly, (72) provides

$$g'(u) - \delta^2/2 \leq 2 \|\nabla f(x)\|_{L^2(\mu)}^2 / (u - L)^2 - \delta^2/2 \leq \|\nabla f(x)\|_{L^2(\mu)}^2 / (16L^2) - \delta^2/2 \leq 0$$

Hence, (111) holds and the output $\lambda^*$ of Algorithm 3 obeys (110) with probability $\gamma$.

It remains to compute a bound on the number of samples from $\mu$ which are required by this procedure. Clearly, by the definition of Algorithm 3, at most $\lceil \log_2(B(u - l)(4 + L + 1)/\epsilon) \rceil$ calls are made to the supergradient oracle given by Proposition 4. Via (88), this yields that at most

$$\frac{(8(4 + L + 1)(\log_2(B(u - l)(4 + L + 1)/\epsilon) + 1))^2 \|\nabla f(x)\|_{L^4(\mu)}^4}{(1 - \gamma)\epsilon^2} \tag{112}$$

invocations of Algorithm 2 are performed with an error parameter which is at least $\epsilon/(2(4 + L + 1)(\max(u - L, 1)))$. Since each invocation of Algorithm 2 requires a single sample of $\mu$, it follows from (112) that

$$\widetilde{O}\left(\frac{L^2 \|\nabla f(x)\|_{L^4(\mu)}^4}{(1 - \gamma)\epsilon^2}\right)$$

samples are used by Algorithm 3– where $\widetilde{O}$ suppresses logarithmic factors in $L, C, M, \|\nabla f(x)\|_{L^2(\mu)}^2$, and $\epsilon$.

Finally, to compute a bound on the expected number of gradient evaluations of $f$ that are performed notice that, for an error parameter of $\epsilon$, each of the $k$ calls to Algorithm 2 (with error tolerance $\epsilon/(4(u - l))$) executes at most

$$t = \max\left(\left\lceil 4\kappa \log\left(\frac{48\kappa \|\nabla f(x)\| (u - l)}{\epsilon}\right)\right\rceil, 0\right)$$

gradient evaluations of $f$; $x$ and $\kappa$ are the random sample and condition number, respectively, which are used in Algorithm 2. Both $x$ and $\kappa$ are random variables, but $\kappa = ((\lambda + L)/(\lambda - \rho_*))^{1/2} \leq (1 + 2L)^{1/2}$ and (due to Jensen)

$$\mathbb{E}\left[\max\left(\log\left(z\right), 0\right)\right] \leq \log \mathbb{E}\left[\max\left(z, 1\right)\right]$$

for any non-negative random variable $z$. Hence, the expected number of gradient evaluations performed by Algorithm 2 obeys the bound

$$\mathbb{E}_\mu\left[t\right] \leq 4\left(1 + 2L\right)^{1/2} \log \mathbb{E}_\mu\left[\left(\max\left(\frac{48(1 + 2L)^{1/2} \left\|\nabla f(x)\right\| (u - l)}{\epsilon}, 1\right)\right)\right] \qquad (113)$$

Summing over the $k$ calls to Algorithm 2 and using the identity $u \leq l + 4L$, one obtains

$$\mathbb{E}_\mu\left[t\right] = \widetilde{O}\left(\frac{L^{5/2}\left\|\nabla f(x)\right\|_{L^4(\mu)}^4}{(1 - \gamma)\epsilon^2}\right)$$

$\square$