# OpenReview forum: "Modified Frank Wolfe in Probability Space"
_NeurIPS.cc/2021/Conference — NeurIPS 2021 Poster_

### Official Review · Reviewer_pe3w · 2021-07-11

**Rating:** 5
**Confidence:** 5

**Summary:**

This paper provides a first-order iterative method based on Local Linear Minimization Oracle (LLMO) to solve smooth optimization problems in probability spaces. It also analyses its convergence rate under a variety of localized relaxation of classical structural assumptions (local Holder smoothness or Lojasiewicz type of inequality) and approximate oracles (gradient and LLMO).

** The answer to the author's answers is in the comment. I raise my score to 5.

**Limitations And Societal Impact:**

yes.

**Main Review:**

The paper originally proposes an algorithm based on LLMO in the context of Probability space. However, I find the paper hard to read with many inaccurate/incomplete statements, missing references, and non-explicit facts preventing the reader from fully appreciating the authors' contribution. A striking example of inaccuracy is that the algorithm is not the "Frank-Wolfe" method adapted to Probability Space. Therefore, I believe that this paper requires a major revision. Hence, although the contribution is appreciable, I strongly recommend rejecting this paper. I now provide a detailed review which will hopefully be helpful for the major revision.

Major comments.


- Algorithm description. Algorithm 3 is a non-standard first order method that iteratively solve linear minimization oracle on a shrinking Wasserstein neighborhood of the current iterates.
The size of the neighborhood is a trade-off between the progression made at each iteration and the computational tractability of the linear minimization subproblems.
The algorithm is itself very interesting but is almost not explained nor compared/related to existing Frank-Wolfe methods.
The explanations are confusing and too much space is attributed to speculative discussion about futur work rather that explaning the algorithm (see, e.g., Line 202-220).\
Also, a first obvious remark is that it should not be called "Frank-Wolfe for (3)" (or the paper be called "Frank-Wolfe on Probability Space"), it is absolutely not the classical form of the original Frank-Wolfe algorithm, for which every subproblems is a linear minimization oracle over the initial constraint domain.\
The algorithm uses  Local Linear Minimization Oracle (LLMO) very reminiscent to the LLMO use in the conditional gradient methods of [Garber2013] and following works. Unfortunately, the authors do not mention any of these previous papers using such LLMO.\
Finally, there are also some notational problems that make it hard to read, e.g. $s$ as the stopping criterion with the norm of the gradient, in (15) $\nu$ should belong to $B_{\mathcal{W}}(\mu_{i-1}, \delta)\cap \mathcal{P}_2(\mathbb{R}^d)$ or the choice of $\delta$ that can be confused with that of (2).

- Theorem 1 is not properly stated. What does an "appropriate choice of input parameters" mean? In Assumption 1, $S$ is not specified to be geodesically convex; is it in Theorem 1? What is the nature of $S$? In the Holder definition, the interplay between $S$ and $\Delta$ could gain from being clarified. Do the authors assume $J$ to be geodesically convex?\
Is $\mu_0$ belonging to $S$? Are there guarantees that (15) maintains the iterates in $S$? The questions are rhetoric here to outline the riddle the reader has to go through to understand the significance of the results.\
Also, Assumption 2 is not discussed in conjunction with $S$. To draw a parallel with the normed space setting, if $S$ is an open neighborhood in the compact constraint set $\mathcal{C}$, vanilla Frank-Wolfe converge linearly when the function is smooth and strongly convex (and the convergence results of FW with Holder-smoothness and Lojasiewicz inequality are discussed in [Ker18]).\
As currently written, it is impossible to understand the significance of Theorem 1. (not to mention that numerical results do not illustrate the theorem).

- Unfair or unclear devaluating of related work.
In "However, [39] narrowly considers the Sinkhorn barycenter problem [...]" the wording "narrowly" is inappropriate and inaccurate. There are seemingly no significant difficulty at extending the results of [Liuse2019] to functional $J$ that satisfy the smoothness assumption proposed in [Liuse2019], i.e., outside the Sinkhorn barycenter problem. Also, example 4.1. relies on [Liuse2019] to establish the Wasserstein smoothness of the objective function. Yet, the authors only mention it in Section 4.
However, it is important to relate it in the introduction since it outlines that a main challenge of optimization in probability spaces is to establish the structural properties satisfied by the problems and not only prove the convergence behavior under these structural properties (the current focus of the authors). This fact appears very clearly in the paper of [Liuse2019], where the main technical challenge seems to establish these structural assumptions rather than extending the FW convergence results to probability spaces.\
Also, it would be better to clarify the distinction between the algorithm in [Liuse2019] and Algorithm 1: indeed, the algorithm of [Liuse2019] is much closer to the "Frank-Wolfe in Probability Spaces". For instance, in [Liuse2019], they solve a similar type of problem with smooth objective function only that they solve the constraint problem on $\mathcal{M}_+(\mathbb{R}^d)$ rather than on $\mathcal{P}_2(\mathbb{R}^d)$. Is that difference important?\
So, the term "narrowly" is confusing and a  bit impolite. Finally, see the list of minor comments for more examples of inadequate treatments of the related works.

- Importance of Assumption 2. From the previous point, if both methods are a Frank-Wolfe on probability space, then how do the $\mathcal{O}(1/T)$ of [Liuse2019] is to be compared with the linear convergence rates of Theorem 1 (when the function is smooth and the Lojasiewicz inequality is global with parameter $\theta=1/2$)? Is the Frank-Wolfe of [Liuse2019] simply less efficient?\
Actually, I tend to believe that Assumption $2$ plays a central role in the distinction. To draw a comparison with known convergence rates of the Frank-Wolfe methods in normed spaces, the type of rates obtained in Theorem 1 corresponds to the settings when the optimum is in the interior of the constraint set. If the comparison holds, then the $1/T$ convergence rate in [Liuse2019] may stem from the fact that "the domain $\mathcal{M}_+^1(\mathcal{X})$ has empty interior in the space of finite signed measure." [Liuse2019] such that the comparison cannot be drawn. In this paper, we do not know to what the rate should be compared: Is the choice of "S" such that the solution is in the interior of $\mathcal{P}_2(\mathcal{X})$? Does it even make sense? What is the cost of an LLMO iteration? Is it comparable to a proximal step or that of an LMO? Overall, it seems very tangential to see Algorithm 1 as a Frank-Wolfe method.

- Comparison of Theorem 1 with known results in "similar" settings. How does the convergence rates compares with known convergence regimes of Frank-Wolfe methods? The authors seems to compare their convergence result with that of proximal methods. Is it because the authors consider the LLMO as kind of proximal steps? This might be the case, since it is not really comparable with a LMO on the initial constraint domain.

- Lojasiewicz conditions. The current presentation of the convergence results with objective functions satisfying Lojasiewicz inequality seems shallow since the class of functions satisfying Loja in that setting is not discussed, no examples are given, and references are missing. I tend to believe that the paper would be clearer without extension to Lojasiewicz inequalities in the main paper but discussion about the meaning of Assumption 2. 1) Lojasiewicz inequalities seem not particularly interesting if they do not come with the theory that explains for which problems' structures they hold generically; 2) missing references about the uses of Lojasiewicz type condition with Frank-Wolfe algorithms, i.e., [Ker2019,Rinaldi2020]; 3) the author present it as a contribution (remark 4) to provide these Lojasiewicz type of inequality, while it is very straightforward or already explored in the context of probability spaces [Bolte2016,Hauer2017]; 4) no examples are given in the numerical experiment part with $\theta\neq 1/2$; 6) no discussion about the nature of the subset $S$ on which it is satisfied.\
For point 1), the crux of the power of Lojasiewicz inequalities is that they are known to hold generically for some classes of functions, e.g., subanalytical functions, see [Bolte2007] for references and extension of the classes of function on which it holds generically. However, the authors do not discuss how this applies in their setting, which I believe can be challenging. Important references exploring these directions are missing, e.g. (but probably others), [Bolte2016,Hauer2017].\
Line 40 "Even in the case where the cost in the squared Euclidean norm $c(x,y)=\|x-y\|^2$ (the case of primary interest [...]". Also, in the numerical experiment, the authors only consider (but do not prove it) strongly convex objective functions. So, the Lojasiewicz extension seems relatively shallow. It is not motivated by practical examples, nor is guaranteed by the theory to hold generically.\

- Numerical Experiments. There are no log plots, and it appears that the convergence is sublinear and not even linear in the asymptotic. So experiments do not support the theory?
The authors do not discuss how the given examples satisfy the Lojasiewicz assumption, and I believe they are only strongly convex.  In line 278, the authors write, "our method reaches the global optima very quickly". This is not very professional; we expect a link with the theoretical results.

- Important references are missing.\
The Kernel Herding methods with Frank-Wolfe [Bach2012]. This is an important reference as it deals with the difficulty of obtaining a faster rate of convergence on Frank-Wolfe in the infinite-dimensional setting. These papers [Bach2012,Lacoste2015] could be discussed in a related work paragraph on Frank-Wolfe methods.\
Missing ref about Lojasiewic in metric spaces, e.g. [Bolte2016,Hauer2017] and others.\
There are specific papers that provide a complete description on the convergence of Frank-Wolfe methods under Holder-smoothness and Lojasiewicz type of inequalities [Ker18,Ker19,Rinaldi2020,Bomze2021].\
Other papers about infinite dimensional FW in probability spaces like [Denoyelle2019,Mensch2019] and others.\
[Garber2013] for Local Linear Minimization Oracle.\
[Guelat1986] for linear convergence of Frank-Wolfe with only smoothness and strong convexity when the optimum is in the interior of the constraint set.\



Others semi-major comments:

- The narrative about the origin of the FW scheme is unclear. Line 29 "development of our FW method is inspired by efforts in DRO [42,5,24,52]". This is confusing because the authors ultimately solve another problem (3). And the confusion is reinforced by the $\delta$ parameter in Algorithm 1 that does not correspond to (2).
It rather seems that [42,5,24,52] arguably inspired the paper to focus on problems such as (2), but where is the link with Frank-Wolfe? How did these works inspire the authors' algorithmic development with the Frank-Wolfe algorithms? It is rather the paper from [Liuse2019] (and [Mensch2019] for which the authors missed the reference although they are indicated in the Liuse paper) that inspired the Frank-Wolfe scheme for that type of problem.\
In Line 32-33, "The form (2), itself, immediately suggests the basis of an infinite-dimensional Frank-Wolfe procedure since it provides a "linear" objective [...]". This explanation is not convincing and looks like an attempt to write another narrative about the origin of using Frank-Wolfe (maybe the reader becomes overly suspicious); in page 4 of [Liuse2019], the comment "[...] in this work we adopt the Frank-Wolfe we adopt the Frank-Wolfe (FW) algorithm. Indeed, one key advantage of this method is that it is formulated in terms of directional derivatives along with feasible directions [...]" is a more convincing reason for using FW algorithms specifically for these types of problems.\
There are also issues with the paragraph "Variational methods" in Line 66. "[...] which draws similar inspiration from finite-dimensional Frank-Wolfe procedures", it is unclear and might lead the reader to believe that the authors deal with the infinite-dimensional setting while other works did not; which is false since e.g. [Liuse2019] already did. The authors do not mention (in the related work paragraph, although they do it in the numerical experiment) also that an important contribution of [Liuse2019] is to prove smoothness of their functional w.r.t. the total variation norm, and the authors use these results in section 4.1.

-  Unclear claims: in Line 53-56, "Unfortunately, the techniques offered by these efforts require assumptions which are too restrictive for this work." For the reader, this statement becomes unclear given the lack of details in Theorem 1, or the authors not discussing when their Lojasiewicz types of inequality are available nor discussing either how restrictive the local richness assumption is.\
In Line 136-138, "To resolve this issue, and to simplify our treatment, this work utilizes the existence of an oracle for the computation of a Wasserstein gradient." The wording "to simplify our treatment" is doubtful: isn't it a necessity (which is perfectly fine but triggers doubt during reading)?
Besides, it is unclear how such an assumption does not limit the result of this work to the scope of the previous works, which were deemed not to be general enough. The authors could expand on that. It is always uncomfortable to read claims about generality with respect to previous works and then see the results with restrictive assumptions that are not discussed.

- First-order methods and Frank-Wolfe. It is relatively well-known that Frank-Wolfe methods have different rates of convergence than other first-order methods. However, in several places, the authors keep on referring to non-projection-free first-order methods, e.g., "we recover sublinear rates that are to be expected from finite-dimensional analogs [34]" (Line 27-28) or "A common approach to establishing improved iteration complexities for convex [...]" (Line 210-211). \
The reference to [34] is strange; they do not deal with Frank-Wolfe algorithms. The linear convergence rates of Frank-Wolfe are sometimes compared with that of other first-order methods, e.g., when the unconstrained optimum is in the interior of the constraint set. An in-depth discussion of this assumption in the context of (2) would be interesting. Several papers seem to suggest that the assumptions that lead to accelerated convergence results on Frank-Wolfe are relatively similar in finite-dimensional or infinite-dimensional settings or in normed spaces versus metric spaces. Here, the authors do not connect with these types of results.

- The title is too generic with respect to the content of the paper. It is also improper; it is far from being a canonical Frank-Wolfe algorithm. The current title is detrimental to previous and future works on the topic.

- The introduction brings some confusion as to what problem is being solved. It is only via the caption of Algorithm 1 that the reader understands that (3) that is being solved.

Minor comments.
- Line 9: "reasonable assumptions"? "mild"?

- Line 20-21: "problems in the form of (1) provide access to rich infinite dimensional structure that sidesteps brittle artifacts of finite dimensional formulations". Could you add references and details of specific examples of "brittle artifacts"?

- Line 36-37 "such works have failed to consider (2) within the context of a general variational method for (1)." the wording "have failed" is violent and unfair.

- Line 62-65, "In that work, smoothness of the objective in (2) is used to, qualitatively, argue that [..]. In contrast, we provide quantification of the level of robustness required to achieve this tractability and demonstrate [...]": here, there is no reference as to where the authors deal with that in the paper; also, the wording "quantitatively" and "demonstrate" are ambiguous: do the authors provide a theoretical result? Or rather provide numerical schemes to better chose the level of smoothness? In the absence of forward reference, it is hard to understand.

- Line 69: "(with quantitative bounds on complexity and convergence)": I would drop "quantitative".

- Line 97 "is a Polish spaces [57]": not defined.

- Line 107-108 "gradients must be defined in terms of a selections in an appropriation cotangent bundle space". Typo on selections. But most importantly, these important technical details are not sufficiently explained. The authors allow space for speculative discussions as in Line 209-220 or long sentences to explain common sense (e.g., Line 48-50).

- Line 107-108 "In particular, as $P_2(R)$ is curved under W (see Appendix A), gradients must be defined in terms of selections in an appropriate cotangent bundle". In Appendix A, the proof of the statement states in Line 602 "The final bullet [the space curvature results] is a direct restatement of the results in Section 8.4. in [1]". So, that would be nice in the main to add (along with the forward reference to the appendix) directly the reference to [1]. Otherwise, it provides the wrong impression that it is the authors' contribution (no one read the appendices).

- Line 109 "is essentially": I don't know what "essentially" refers to in that setting.

- Line 125 "the curious reader": a bit annoying when the reader is already annoyed by the lack of clarity.

- Line 156: "In finite dimensions, iterative, gradient-based ..." There exists infinite-dimensional work already, so why suggesting that this work provide a novel view on infinite dimensional settings?
Also, at first sight, the paper does not describe all gradient based methods but Frank-Wolfe methods, so why keeping it general to first order methods?

- Line 168-178: No mention of the analogous results in normed spaces and no mention of what structure to assume on $J$ to ensure the existence of Lojasiewicz inequality with a non-trivial parameter. This is however the main appeal of Lojasiewicz inequalities.

- Line 180: In algorithm, the second line in the for loop: I found it strange to ask to compute s (and actually, the instruction is even less clear since it asks to compute an inequality which makes no sense), why not just directly taking the norm of your approximate gradient as the stopping criterion?

- Line 180: The choice of $\delta$ in Algorithm 3 is confusing since it could refers to Equation (2) while the algorithm seek to solving Equation (3).

- Line 191-195: Please provide the reference as to where the proof is in the appendix.

- Line 209-220: I find it too speculative; this is the type of discussion that should be in the conclusion. Also, it is at odd to elaborate on a futur results when the algorithm itself has not been properly introduced. Besides, the elaboration concern techniques of acceleration for proximal methods, not conditional gradient methods (that are different, see e.g. corrective version of FW or [Comb2020]), so hardly understandable in that context.

Additional references.

[Bach2012] Francis Bach, Simon Lacoste-Julien, Guillaume Obozinski. On the Equivalence between Herding and Conditional Gradient Algorithms. ICML 2012.

[Bolte2007] Bolte, J.; Daniilidis, A. and Lewis, A. The {\L}ojasiewicz inequality for nonsmooth subanalytic functions with applications to subgradient dynamical systems. 2007.

[Bolte2016] J. Bolte, A. Blanchet. A family of functional inequalities: Lojasiewicz inequalities and displacement convex functions. 2016

[Bomze2020] I Bomze, F Rinaldi, D Zeffiro. Frank-Wolfe and friends: a journey into projection-free first-order optimization methods. 2020

[Comb2020] Combettes, C. and Pokutta, S. (2020). Boosting Frank-Wolfe
by chasing gradients. ICML.

[Denoyelle2019] Denoyelle, Q.; Duval, V.; Peyré, G. and Soubies, E. The sliding Frank-Wolfe algorithm and its application to super-resolution microscopy. 2019.

[Garber2013] Dan Garber, Elad Hazan. A linearly convergent conditional gradient algorithm with applications to online and stochastic optimization.

[Guelat1986] J. GuéLat; P. Marcotte. Some comments on Wolfe's away step.

[Hauer2017] D. Hauer, J. Mazon. Kurdyka-Lojasiewicz-Simon inequality for gradient flows in metric spaces. 2017.

[Ker18] T. Kerdreux, A. d'Aspremont, S. Pokutta. Restarting Frank-Wolfe: Faster Rates Under Hölderian Error Bounds.

[Ker19] T. Kerdreux, A. d'Aspremont, S. Pokutta. Restarting Frank-Wolfe. AISTATS 2019.

[Lacoste2015] S. Lacoste-Julien, F. Lindsten, F. Bach. Sequential Kernel Herding: Frank-Wolfe Optimization for Particle Filtering. ICML 2015.

[Liuse2019] G. Luise, S. Salzo, M. Pontil and C. Ciliberto. Sinkhorn barycenters with free support via frank-wolfe methods. NeurIPS 2019.

[Mensch2019] Arthur Mensch, Mathieu Blondel, Gabriel Peyré. Geometric Losses for Distributional Learning. ICML 2020.

[Rinaldi2020] F. Rinaldi, D. Zeffiro. A unifying framework for the analysis of projection-free first-order methods under a sufficient slope condition.

**Time Spent Reviewing:**

20

---

> ### Author Response · Authors · 2021-08-10
> **Author's Response to Reviewer pe3w**
>
> Thank you for your comments, hopefully the following discussion can clear up your concerns. First of all, we would like to apologize if we have offended the reviewer in the context of this work. Specifically we apologize for our poor choice of words when characterizing the scope of [1] as "narrow". This was not meant to belittle the contributions of [1]. It was simply meant distinguish our work from [1] by emphasizing that [1] focused primarily on the Sinkhorn barycenter problem. We would like to respectfully emphasize that our problem setting and analysis techniques are quiet different from that displayed in [1] (the reference which is mentioned by the reviewer many times). Thus, it is very difficult to make comparison between these papers in terms of assumptions and theoretical analysis.
> Here we summarize key differences,
> - The smoothness condition of the "first-order variation" in [1] appears to be induced by the total variation distance in the **general Banach space**. We believe that the distance (i.e. geometry) is a modeling choice which has consequences in terms of the types of applications that can benefit from a such a choice. For instance, total variation is a strong metric that is potentially most compelling when studying functions of probability measures which depend significantly on density characteristics (e.g. the mode of a density, for instance). Alternatively, we exploit the Wasserstein geometry to provide a notion of **Wasserstein differentiability** on a subset of $M_1^+(\mathbb{R}^d)$. In making this choice, we balanced tractability of the analysis and fidelity of the procedure (since the algorithm itself requires a much weaker notion of differentiability to be applicable, namely, Gateaux differentiability). This choice is part of our contributions in this paper, also acknowledged by **reeviewer 8BEc**.
> - After carefully reading reference [1] again, we still are not able to see how the algorithms [1] can be directly applied in our setting. The paper [1] solves the constrained problem on $M_1^+(\mathcal{X})$ (i.e., $\mathcal{X}$) under compact support assumptions. In contrast, we consider  $\mathcal{P}_2(\mathbb{R}^{d})$ without the need of compactness-- we have $\mathcal{P}_2(\mathbb{R}^{d}) \subseteq M_1^+\left( \mathbb{R}^d \right)$. Even under the weak topologies, $M_1^+\left( \mathbb{R}^d \right)$ and $M_1^+\left( \mathcal{X} \right)$ (for compact $X$) are different in a non-trivial way.  Actually, we discuss, below, why, in the absence of compactness, a direct interpretation of the Frank-Wolfe algorithm in the space of probabilities is not well defined, in general.
>
> We believe that there might be some misunderstanding of our contributions (compared with [1]). We hope that the differences that we emphasize may help the reviewer to re-evaluate our contributions. We believe the proposed method could bring a different and new perspective for solving the structured optimization problem over the probability space.
>
> ## Relationship to the Frank-Wolfe algorithm:
>
> >Algorithm description. Algorithm 3 is a non-standard first order method that iteratively solves a linear minimization oracle on a shrinking Wasserstein neighborhood of the current iterates... The algorithm is itself very interesting but is almost not explained nor compared/related to existing Frank-Wolfe methods....
>
> Thanks for pointing out that we need to clarify this. We believe it would
> reduce confusion by changing references to the method of the paper from
> "Frank-Wolfe'' to "modified Frank-Wolfe''. We will update our paper based on your suggestions. This description would also be closer to terminology such as "regularized Frank-Wolfe'' [4] which highlights a connection to Frank-Wolfe but also reflects a non-trivial modification.
>
> We also want to emphasize here that the vanilla Frank-Wolfe method cannot work in the probability space at the generality that we consider in our paper without a natural modification as we do (i.e. introducing a tractable local linear constraint). This is because the (planar/Gateaux) derivative (which corresponds to the influence function) will often be unbounded when the distributions do not have compact support. So, the minimization problem over the space of probability measures which form the basis of the iterates in a direct Frank-Wolfe interpretation often will not be well defined.
>
> It is also our understanding that previous work [1], appearing in this conference, has yielded the moniker of a "Frank-Wolfe'' algorithm on the basis of maintaining iterates which minimize a linear functional over the space of probability measures $M_1^+(X)$ on a compact subset $X \subseteq \mathbb{R}^d$ (see page 5, paragraph 1 in [1]). This is a convex, (weakly) compact subset
> of the space of all finite Borel measures on $X$ and, for the functional
> considered in that paper (the barycenter functional), this set constitutes the
> effective domain of the problem. In comparison, our algorithm maintains iterates
> in $\mathcal{P}_2(\mathbb{R}^{d}) \subseteq M_1^+\left( \mathbb{R}^d \right)$ which minimize a
> linear functional, subject to an additional constraint on the Wasserstein
> distance between the iterates. Since $\mathcal{P}_2\left( \mathbb{R}^d \right)$ is itself
> convex, but not (weakly) compact subset of $M_1^+\left( \mathbb{R}^d \right)$, we note that, with the addition of this "trust-region" constraint (which provides a weakly compact set), our method fulfills the
> ``Frank-Wolfe'' criterion met by [1]. As pointed out by the reviewer,
> the addition of this trust-region constraint is very similar to
> finite-dimensional Frank-Wolfe algorithms which use a Local Linear Minimization
> Oracle (LLMO) [3]. We believe this connection is important, an
> oversight on our part, and we will update our paper to reflect this connection.
> However, this comparison further highlights the justification of our algorithm
> being a "modified Frank-Wolfe'' algorithm. Indeed since, formally, minimization
> of a linear function over a ball of a fixed size is related to minimization
> of a linear function plus a suitable penalty term, we believe that
> parallels can also be drawn to "regularized Frank-Wolfe'' [4]--
> particularly in a way the terminology "modified Frank-Wolfe'' would carry an
> interpretable meaning for our algorithm.
>
> Beyond such comparison-based discussions, we also believe that our algorithm is
> similar in spirit to Frank-Wolfe. Specifically, the algorithm uses special
> structure and geometry of the convex subset $\mathcal{P}_2\left(\mathbb{R}^d\right)$ (coupled
> with the tractability of minimizing a linear functional over this set) to
> produce a first-order descent procedure. Moreover, the use of
> $S \subseteq \mathcal{P}_2(\mathbb{R}^d)$ (see Assumptions 1, 2, and 3) allows the constraint
> set to be further narrowed when $S$ is geodesically convex.

---

> > ### Author Response · Authors · 2021-08-10
> > **Continued response to Reviewer pe3w**
> >
> > ## Issues regarding the clarity of Theorem 1:
> >
> > > Theorem 1 is not properly stated. What does an "appropriate choice of input parameters" mean?
> >
> > This stylistic choice was made to keep the statement of the theorem free from large algebraic expressions which detail how the parameters of the algorithm must be selected to provide the desired convergence guarantees. These parameters are stated in the proof of Theorem 1; see also remark 4.
> >
> > > What is the nature of $S$? In the Holder definition, the interplay between $S$ and $\Delta$ could gain from being clarified. Do the authors assume $J$ to be geodesically convex? Is $\mu_0$ belonging to $S$? Are there guarantees that (15) maintains the iterates in $S$?
> >
> > We believe this to be an area where the paper can be improved, since questions regarding the nature of $S$ were raised by both **reviewer 8BEc** and **reviewer cwwb**. We have proposed changes to the paper to remedy this issue, please see the responses to these reviewers. We believe the only considerations not addressed in those responses would be the following. The distribution $\mu_0$ is assumed to lie in $S$, we will update the language of the paper to reflect this. The use of $\Delta$ in the Holder condition allows smoothness to hold only locally and not depend on global behavior. This condition is at least as general as standard Holder smoothness.
> >
> > > and the convergence results of FW with Holder-smoothness and Lojasiewicz inequality are discussed in [Ker18]
> >
> > We require further clarification from the reader on this point. We believe it is clear from the statement of Assumptions 1-3 that geodesic convexity in $J$ is not required. Thus, it is unclear how [Ker18] or [Ker19] should be made to compare to our analysis, given that convexity assumptions are made in these papers. Also, our analysis uses non-trivial properties of Wasserstein geometry, see our response to **reviewer cwwb** under the heading **Q2: Novelty of results relative to analysis of steepest descent**. Thus, we believe that one cannot simply reproduce our results from the analysis of these papers.
> >
> > ## Issues regarding the charaterization of [1]
> >
> > >Unfair or unclear devaluating of related work. In "However, [39] narrowly considers the Sinkhorn barycenter problem [...]" the wording "narrowly" is inappropriate and inaccurate.
> >
> > We apologize for this language and, as pointed out in the introduction to this response, we will change this language in the paper. The objective of this sentence is merely to observe that [1] focuses on a specific problem, which we do not do in this work.
> >
> > >There are seemingly no significant difficulty at extending the results of [Liuse2019] to functional  that satisfy the smoothness assumption proposed in [Liuse2019], i.e., outside the Sinkhorn barycenter problem.
> >
> > Respectfully, we do not agree with this statement based upon the reasons stated in the introduction to this response. The assumptions and ambient spaces on which these methods operate are fundamentally different. We invite the reviewer to provide further details on how the extension of [1] to our setting would operate.
> >
> > > Also, example 4.1. relies on [Liuse2019] to establish the Wasserstein smoothness of the objective function.
> >
> > We do not believe that this is not a correct charaterization of any statement made in Section 4. The definition of smoothness in [1] and this paper are fundamentally different.
> >
> > > However, it is important to relate it in the introduction since it outlines that a main challenge of optimization in probability spaces is to establish the structural properties satisfied by the problems and not only prove the convergence behavior under these structural properties (the current focus of the authors).
> >
> > We believe that both the tasks of establishing methods and structural properties for infinite dimensional problems are important. Often, they are not simultaneously addressed in the same work and establishing a structural property for an infinite dimensional problem can be, itself, a major contribution. Moreover, the properties of smoothness and {\L}ojasiewicz inequalities are canoical conditions on infinite dimensional functionals-- references are provided in Remark 3, but we can provide additional references if desired.
> >
> > ## Importance of Assumption 2:
> >
> > > Importance of Assumption 2. From the previous point, if both methods are a Frank-Wolfe on probability space, then how do the $O(1/T)$ of [Liuse2019] is to be compared with the linear convergence rates of Theorem 1 (when the function is smooth and the Lojasiewicz inequality is global with parameter $\theta = 1/2$)? Is the Frank-Wolfe of [Liuse2019] simply less efficient?
> >
> > As previously stated we do not believe that the convergence rates of the algorithms are comparable, given the difference in the conditions and settings between the two works. We believe the importance of Assumption 2 is highlighted by our response to **reviewer cwwb** under the heading **Q5: The geodesically convex set $S$**. We have also proposed changes in that response to clarify this issue.
> >
> > > What is the cost of an LLMO iteration? Is it comparable to a proximal step or that of an LMO?
> >
> > The cost of (15), what we believe the reviewer calls a LLMO iteration, is given in Proposition 1. We do not believe that the cost should be compared to the cost of a LMO iteration in finite dimensions simply because the problems which they are solving are fundamentally different. Indeed, what does the reviewer mean by an LMO iteration? Is this an LMO iteration subject to a finite set of linear constraints?
> >
> > ## Lojasiewicz conditions.
> >
> > > Lojasiewicz conditions. The current presentation of the convergence results with objective functions satisfying Lojasiewicz inequality seems shallow since the class of functions satisfying Loja in that setting is not discussed...
> >
> > We believe that {\L}ojasiewicz-type inequalities are fairly standard in the infinite dimensional literature and examples of functionals for which these inequalities are known are provided in the references in Remark 3. Moreover, we believe it is generally accepted that this is a highly relevant condition for descent methods in infinite dimensional settings due trivial examples where a form of geodesic convexity (with respect to Wasserstein distance) fails to hold, but a {\L}ojasiewicz inequality is still satisfied. Indeed, a simple example is Wasserstein distance itself. It is well-known [Chapter 7, 2] that Wasserstein distance fails to be strongly geodesically convex with respect to itself while it satisfies the {\L}ojasiewicz-type inequality that would be implied by strong geodesic convexity-- this is a trivial result of the properties listed in Appendix A. Moreover, we believe it is be clear that geodesic convexity for a functional with a Wasserstein-bounded sub-level set implies a {\L}ojasiewicz inequality-- see Remark 3. Thus, the set of functionals satisfying these inequalites is at least as general as geodesically convex functionals with bounded sub-level sets. We are happy to further modify Remark 3 to clarify this. We are also happy to add the references cited by the reviewer ([Bolte2016,Hauer2017]) in our discussion of these inequalites.
> >
> > ## Issues relating to numerical experiments:
> >
> > > Numerical Experiments. There are no log plots, and it appears that the convergence is sublinear and not even linear in the asymptotic. So experiments do not support the theory?
> >
> > Please see our response to **reviewer cwwb** under the heading **Q3: Regarding the experiments** for details regarding the computational examples and the assumptions of the paper. We will reiterate, however, that the objective of this section is not to redo the theory of the previous sections but, rather to demonstrate practical relevance. We would also caution the reviewer against expecting an asymptotically decreasing residual given that our method is implemented using both an inexact oracle for (15) and an inexact gradient oracle.
> >
> > ## Issues relating to missing references:
> >
> > > The Kernel Herding methods with Frank-Wolfe [Bach2012]. This is an important reference as it deals with the difficulty of obtaining a faster rate of convergence on Frank-Wolfe in the infinite-dimensional setting. These papers [Bach2012,Lacoste2015] could be discussed in a related work paragraph on Frank-Wolfe methods....
> >
> > We believe the setting and assumptions of [Bach2012] are fundamentally different from ours. Would the author please clarify the mapping from the setting of that work to our work? We are happy to add references [Ker18,Ker19,Rinaldi2020,Bomze2021] to give the reader a wider scope of Frank-Wolfe methods and simply mention that the assumptions are fundamentally different. However, a detailed discussion would be, we believe, too distracting especially given the space limitations.
> >
> > ## References
> > [1] Luise, Giulia, et al. "Sinkhorn Barycenters with Free Support via Frank-Wolfe Algorithm." Advances in Neural Information Processing Systems 32 (2019): 9322-9333.
> >
> > [2] Ambrosio, Luigi, Nicola Gigli, and Giuseppe Savaré. Gradient flows: in metric spaces and in the space of probability measures. Springer Science \& Business Media, 2008.
> >
> > [3] Garber, Dan, and Elad Hazan. "A linearly convergent variant of the conditional gradient algorithm under strong convexity, with applications to online and stochastic optimization." SIAM Journal on Optimization 26.3 (2016): 1493-1528.
> >
> > [4] Migdalas, Athanasios. "A regularization of the Frank—Wolfe method and unification of certain nonlinear programming methods." Mathematical Programming 65.1 (1994): 331-345.

---

> > > ### Comment · Reviewer_pe3w · 2021-08-31
> > > **response to authors' answer**
> > >
> > > The explanations given by the authors are very insightful and interesting.
> > > In particular, the difference between the Liuse et al. and their setting is now well explained.
> > > However, I still believe that the paper needs a lot of changes to best impact the community. Hence, because many of the authors' answers are clarifying many points, I raise my score to 5.
> > >
> > >
> > > However, the paper still requires a major revision which may not fit with the purpose of the camera-ready phase.
> > > Here is a non-exhaustive list of some points that are not really addressed or that highlight the magnitude of the changes required.
> > >
> > > 1) As I pointed out in my review, one major reason making this paper hard to follow is the link with Frank-Wolfe. In the authors' answer, I only found one explanation convincing to justify the use of Frank-Wolfe: i.e., when answering reviewer cwwb:
> > > "Specifically, the algorithm uses special structure and geometry of the convex subset
> > > (coupled with the tractability of minimizing a linear functional over this set) to produce a first-order descent procedure."
> > > But I still have several concerns:
> > > - What do you mean by the geometry of the convex subset S apart from being geodesically convex?
> > > - Although the authors' answer clarify the reason for considering LLMO, it still seems very confusing. i) it is a major change to the paper to add these explanations; ii) the paper on Modified FW is much less known than LLMO or the steepest descent method [Boyd, 2004] with the Wasserstein distance as pointed out by reviewer cwwb; iii) the reviewer did not address my concern about the speculative paragraph explaining how to extend their Algorithm. The explanation draws on analogies with non-Frank-Wolfe first-order methods as if the best way to consider the algorithm was not really as a Frank-Wolfe one (which I still believe). -- Please note that the references I provided in my first review are indeed not relevant if the analogy with Frank-Wolfe methods in general Banach Spaces does not hold.
> > >
> > >
> > > 2) It is a significant change to the paper to explain that the numerical experiments do not aim at supporting the theoretical results.
> > > "We would also caution the reviewer against expecting an asymptotically decreasing residual given that our method is implemented using both an inexact oracle for (15) and an inexact gradient oracle." Unless stated otherwise in the paper, I think that a reader first expects the experiments to reflect the theory.
> > > I am not saying that it is necessarily a problem if the numerical experiment could not illustrate the example, but it has to be made very clear, which is not the case here.
> > >
> > > 3) Lojasiewicz:
> > > In the paper remark 3, I read: "More restrictive versions of both (13) and (14) commonly appear in previous literature [3, 31, 37, 18, 14]" (where (13) is the Holder-smoothness assumption and (14) is the Lojasiewicz inequality).
> > > And in the review I read: "We believe that {\L}ojasiewicz-type inequalities are fairly standard in the infinite dimensional literature and examples of functionals for which these inequalities are known are provided in the references in Remark 3."
> > >
> > > I also believe that {\L}ojasiewicz-type inequalities are standards. But reading your remark 3, the sentence "More restrictive versions of both (13) and (14)" and "Additionally, statement of the Łojasiewicz inequality (14) is broader than canonical treatments due to the presence of the auxiliary power $\theta$." seems incorrect. I read these sentences as a manner to claim some novelty with that respect which I think is incorrect. The sentence "We are also happy to add the references cited by the reviewer ([Bolte2016,Hauer2017]) in our discussion of these inequalites" is also strange. It rather seems that you missed these references that have already considered statements like (14) in a probability space.
> > >
> > > Another point that is "fairly standard" in the infinite-dimensional literature is that the interest of Lojasiewicz inequality is that they are known to hold *generically* for classes of functions which was the crux of my question in the review. I find the answer relatively satisfying, but I insist that such a discussion is not in remark 3 and is fundamental to add.

---

> > > > ### Author Response · Authors · 2021-09-01
> > > > **Response to reviewer pe3w's comments.**
> > > >
> > > > We would like to thank reviewer pe3w for their further feedback. Such continued dialogue is, in our opinion, the heart of the peer-review process.
> > > >
> > > > We are glad to see that major concerns have been addressed and we thank you for raising your score.
> > > >
> > > > With regard to the changes (which we have proposed) that address these concerns, we believe that all of these changes can be easily made during the camera-ready phase. The changes are mostly commentary and do not change the algorithm, the assumptions, convergence results or experiments. Moreover, nearly two months are available to prepare the revision and the paper will be allowed an additional content page for the camera-ready version. We strongly believe that our proposed changes will constitute less than an additional page of commentary and, therefore, can be easily added without major restructuring or substantial removal of material.
> > > >
> > > > More importantly, we believe that the adjustments to the paper are minor. For instance, clarifying why a modification of the Frank-Wolfe method is required (in the general probability space that we consider) is immediate after recognizing that the first variation (i.e. planar derivative) could, easily, be unbounded. In what follows, we provide the precise modification plan based on your comments.
> > > >
> > > > ## Modification incorporating LLMO
> > > >
> > > > > Although the authors' answer clarify the reason for considering LLMO, it still seems very confusing. i) it is a major change to the paper to add these explanations;...
> > > >
> > > > We believe that it is not a hard, nor a drastic change, to clarify the reason for considering LLMO. Here are two major points we will add to our paper to better motivate LLMO.
> > > >
> > > > - (Motivation) As we mentioned in the earlier response, the vanilla Frank-Wolfe method cannot work in the probability space in general, as the planar derivative can be unbounded when, say, a distribution does not have compact support. Hence, the modified Frank-Wolfe step is necessary.
> > > > - (Tractability) By localizing the problem to a Wasserstein ball, the natural modification that we make in this paper not only makes the vanilla Frank-Wolfe problem sensible, but it also renders the problem computationally tractable via finite dimensional convex optimization.
> > > >
> > > > ## Modification clarifying experimental results
> > > >
> > > > > It is a significant change to the paper to explain that the numerical experiments do not aim at supporting the theoretical results. ...
> > > >
> > > > Respectfully, we disagree that this additional explanation constitutes a major change since it does not alter any claim originally made in the work. Indeed, let us reiterate the changes that we proposed in our response to *reviewer cwwb* (under the heading *"Q3: Regarding the experiments"*)-- hopefully this illustrates that these changes are purely commentary, adding only to exposition and not modifying any previous claims.
> > > >
> > > > - A concise summary of extent to which smoothness and PL inequalities are known for the computational examples.
> > > > - A clarification that the purpose of the computational examples is not to redo the theoretical analysis of the paper, but to demonstrate practical relevance of the proposed algorithm.
> > > >
> > > > ## Minor remaining issues
> > > >
> > > > The reviewer also provided several, remaining (minor) concerns that we believe are easily addressed .
> > > >
> > > > > the reviewer did not address my concern about the speculative paragraph explaining how to extend their Algorithm. The explanation draws on analogies with non-Frank-Wolfe first-order methods as if the best way to consider the algorithm was not really as a Frank-Wolfe...
> > > >
> > > > We are willing to remove the paragraphs between lines 202-220 if the reviewer believes that: 1) this discussion is too speculative or 2) this discussion confuses the reader about the relationship of the algorithm to the Frank-Wolfe method and it's modifications - particularly if this removal would, in a material way, further effect the reviewer's assessment of the work. Alternatively, we would also be willing to modify the language in these paragraphs in the following manner. The primary purpose of this language is to demonstrate why averaging of iterates (a basic technique common to many acceleration schemes in finite dimensions) is difficult to apply to obtain further improvements to our algorithm. This technique is not just key for the acceleration of proximal methods, but it is also useful for the acceleration of Frank-Wolfe methods (such as in [1], which is brought up by the reviewer). The language in these paragraphs could be (easily and simply) modified to clarify this and highlight that the discussion encompasses a technique which is also used to accelerate Frank-Wolfe methods (not just proximal methods). Further, if it further helped with the flow or exposition of the work, we would be willing to move this discussion to the conclusion section. In any case, once again, this is not a major modification in the paper since the main contributions are not affected.
> > > >
> > > > > I read these sentences as a manner to claim some novelty with that respect which I think is incorrect. The sentence "We are also happy to add the references cited by the reviewer ([Bolte2016,Hauer2017]) in our discussion of these inequalites" is also strange. It rather seems that you missed these references that have already considered statements like (14) in a probability space.
> > > >
> > > > We note that "[Hauer2017]" appears as reference "[31]" in our paper and is cited in the paragraph in question (Remark 3); we apologize for not bringing this up in our original response.
> > > > However, since the formulation of these inequalities per-se does not constitute contribution of the paper, we believe that neither [Bolte2016] nor [Hauer2017] is central to the contributions of this paper. The references are relevant to position the nature of the conditions assumed and surely, there are always non-centrally-important references that can be added to a work in order to better inform the reader. However, faulting a work for a lack of such a reference might, perhaps, be a bit onerous; with 65 total relevant references to various topics well-connected to this paper, we believe that this work reflects a good faith effort to provide an appropriate (but non-exhaustive) account of related literature.
> > > >
> > > > Should it clarify the paper, and materially impact the reviewer's assessment of the work, we would also be willing to add a sentence to Remark 3 which explains the following. We are not claiming novelty for the mere formulation of Holder smoothness and a Łojasiewicz inequality. Rather, the use of both of these conditions (particularly with the presence of the auxiliary powers $\alpha$ and $\theta$) to provide concrete convergence rates for a computationally-implemented, infinite-dimensional descent method does not appear in related literature as far as we know.
> > > >
> > > > > What do you mean by the geometry of the convex subset $S$ apart from being geodesically convex?
> > > >
> > > > For the analysis of the algorithm, beyond geodesic convexity, a subset $S \subseteq \mathcal{P}_2(\mathbb{R}^d)$ has a natural "differentiable/smooth" geometry. The quoted terms are given rigorous meaning via the statements of Proposition 2. Formally, this geometry allows members of $S$ to be related to one another (in a "differentiable" way) using dual objects.
> > > >
> > > > Thank you so much, again, for your time and effort to make our work better.
> > > >
> > > > [1] Combettes, C. and Pokutta, S. (2020). Boosting Frank-Wolfe by chasing gradients. ICML.

---

### Official Review · Reviewer_cwwb · 2021-07-19

**Rating:** 5
**Confidence:** 2

**Summary:**

This paper proposes an algorithm for the minimization of functionals of probability distributions. The authors derive a convergence rate in the case of a smooth and PL function (under the geometry given by the Wasserstein distance of order 2). They also run some synthetic experiments of non-parametric estimation

**Limitations And Societal Impact:**

Yes

**Main Review:**

## Main Points
### Originality and significance:
I am not an expert in the area of Distributionally Robust Optimization (DRO) and Variational method which are the areas addressed in the related work of this paper. However, I can comment on the originality of this paper with respect to the optimization literature.
It seems to me that the connection between the main algorithm (alg. 1) and the Frank-Wolfe algorithm is not clear. Algorithm 1 seems to be a steepest descent method [Boyd, 2004] with the Wasserstein distance: at each timestep an estimation of the gradient is computed and then a minimizer of the inner product with the gradient within a delta ball is computed in (15).

Moreover, the Frank-Wolfe algorithm is a constrained convex optimization method. The problem considered in (1) is unconstrained. It is not clear how the set S is handled and if S is actually a constraint or not. All this to say that I do not see any clear connection with the Frank-Wolfe algorithm. Can the authors comment on the following question: to what extent your algorithm is related to Frank-Wolfe and steepest descent method? What is the novelty of your analysis with regards to a standard analysis of steepest descent method under PL and smoothness assumptions?

### Clarity:
The topic of this paper is quite technical and I think this paper does an OK job at introducing the notions. However, one missing key point in this paper is the algorithm to compute the approximate gradient oracle (called ‘Frank Wolfe step’ in the paper.. Though it is more a steepest descent step in my opinion). I looked at the appendix quite extensively and it seems that one should run algorithm 3 to get a $\lambda$, sample $x \sim \mu$  and then return $m_\lambda(x)$ see (99). It would be useful to have a pseudo algorithm summarizing this in the main paper.

### Soundness of the experiments:
Are the experiments within the assumptions? It is not mentioned or justified that the functions considered are smooth and PL.
A significant part of the related work is dedicated to DRO? Since this work cannot handle constrained optimization (J has to be smooth) how is this work related to DRO?
About the Set S: All your assumptions mention a set $S$: If the set $S$ is not the whole space, how do you make sure that the iterates stay in $S$? What are non-trivial examples of sets $S$?



## Minor details:
- L29: DRO in mentioned before being defined
- L129 Gauteaux
- L135 What is ‘the supremum”
- L660 a ‘\hat’ is missing
- Eq (54), (65) and (58) it seems that a square is missing on $g_{i-1}$ (thankfully since you need it for the second line of (58))

## References:
Boyd, Stephen, Stephen P. Boyd, and Lieven Vandenberghe. Convex optimization. Cambridge university press, 2004.

### I have read the authors' responses and the other reviews. I stand with the points of reviewer pe3w so I will maintain my score.

**Time Spent Reviewing:**

4

---

> ### Author Response · Authors · 2021-08-10
> **Author's Response to Reviewer cwwb**
>
> Thank you for taking the time to review our paper and for your insightful comments. We find that they are very constructive and we hope to clear up your lingering concerns with the following responses.
>
> ## Q1: Relationship to the Frank-Wolfe algorithm
>
> > Can the authors comment on the following question: to what extent your algorithm is related to Frank-Wolfe...
>
> This comment highlights the need for us to add a remark to the
> work which clarifies it's relationship with traditional Frank-Wolfe approaches-- we will do this based upon all of the reviewers' feedback. Further, since our method is not described by a literal, verbatim translation of the conditional gradient method (from finite dimensions) we believe it would reduce confusion by changing references to the method of the paper from
> "Frank-Wolfe'' to "modified Frank-Wolfe''. This description
> would also be closer to terminology such as "regularized Frank-Wolfe''
> [7] which highlights a connection to Frank-Wolfe but also reflects a
> non-trivial modification. We believe, however, that such modification is necessary if a reasonable extension of the Frank-Wolfe procedure is to be considered over the space of probability measures. The vanilla Frank-Wolfe method cannot work in the probability space at the generality that we consider in our paper without a natural modification as we do (i.e. introducing a tractable local linear constraint). This is because the (planar) derivative (which corresponds to the influence function) will often be unbounded when the distributions do not have compact support. So, the minimization problem over the space of probability measures often will not be well defined.
>
> It is also our understanding that previous work [8], appearing in
> this conference, has yielded the moniker of a "Frank-Wolfe'' algorithm on the
> basis of maintaining iterates which minimize a linear functional over the space
> of probability measures $M_1^+(X)$ on a compact subset $X \subseteq \mathbb{R}^d$ (see
> page 5, paragraph 1 in [8]). This is a convex, (weakly) compact subset
> of the space of all finite Borel measures on $X$ and, for the functional
> considered in that paper (the barycenter functional), this set constitutes the
> effective domain of the problem. In comparison, our algorithm maintains iterates
> in $\mathcal{P}_2\left(\mathbb{R}^d\right) \subseteq M_1^+\left( \mathbb{R}^d \right)$ which minimize a linear functional, subject to an additional constraint on the Wasserstein
> distance between the iterates. Since $\mathcal{P}_2\left( \mathbb{R}^d \right)$ is itself
> convex, but not (weakly) compact subset of $M_1^+\left( \mathbb{R}^d \right)$, we note that, with the addition of this "trust-region" constraint (which provides a weakly compact set), our method fulfills the
> ``Frank-Wolfe'' criterion met by [8]. As pointed out by a reviewer,
> the addition of this trust-region constraint is similar to
> finite-dimensional Frank-Wolfe algorithms which use a Local Linear Minimization
> Oracle (LLMO) [9]. We believe this similarity is important and we will update our paper to reflect this connection.
>
> However, this comparison further highlights the justification of our algorithm
> being a "modified Frank-Wolfe'' algorithm. Indeed since, formally, minimization
> of a linear function over a ball of a fixed size is related to minimization
> of a linear function plus a suitable penalty term, we believe that
> parallels can also be drawn to "regularized Frank-Wolfe'' [7]--
> particularly in a way the terminology ``modified Frank-Wolfe'' would carry an
> interpretable meaning for our algorithm.
>
> Beyond such comparison-based discussions, we also believe that our algorithm is
> similar in spirit to Frank-Wolfe. Specifically, the algorithm uses special
> structure and geometry of the convex subset $\mathbb{P}_2\left(\mathbb{R}^d\right)$ (coupled
> with the tractability of minimizing a linear functional over this set) to
> produce a first-order descent procedure. Moreover, the use of
> $S \subseteq \mathbb{P}_2(\mathbb{R}^d)$ (see Assumptions 1, 2, and 3) allows the constraint
> set to be further narrowed, so long as $S$ is geodesically convex.
>
> ## Q2: Novelty of results relative to the analysis of steepest descent
>
> > What is the novelty of your analysis with regards to a standard analysis of steepest descent method under PL and smoothness assumptions?
>
> We may require further clarification from the reviewer in order to answer this question appropriately. However, our interpretation of this question is: *if one merely takes Definitions 1-4 and Assumptions 1-3, is it possible to simply rearrange these conditions to construct a proof of Theorem 1 which, nearly
> verbatim, mirrors a proof of convergence for steepest descent under analogous conditions (on $\mathbb{R}^d$)?*
>
> This answer to this question is no and this is perhaps best illustrated by our response to the question of **reviewer 8BEc** regarding whether our convergence results would directly follow through for Wasserstein distance $W_p$ with $p \neq 2$. As illustrated by that discussion, specific properties of Wasserstein geometry for $p=2$ (connections between Kantorovich potentials and convex functions, kinematic descriptions of geodesics, etc.) are used to produce Theorem 1 --- these properties are not necessary and/or do not have analogues in $\mathbb{R}^d$. Thus, the novelty of our analysis lies in showing how the geometry of the Wasserstein metric can be integrated in a non-trivial way with rather weak smoothness assumptions and a rather general statement of a {\L}ojasiewicz
> condition to produce a first order method that recovers convergence rates which are suggested, but not implied, by certain canonical analyses of finite-dimensional, steepest decent methods.
>
> Indeed, supposing that the reviewer intended a relatively strict comparison with [1], we would add that our analysis also incorporates the use of an inexact step. Further, we demonstrate how to construct an oracle to compute this approximate step (Proposition 1) - this task being non-trivial. Precisely, prior to this work it was not known that (15) could be computed choices of $\delta$ which can be explicitly quantified and are relatively ``large" in the sense of being uniformly bounded away from zero over a class of expectations of interest. The lower bound on $\delta$ would be large enough to permit the implementation of a first order procedure similar to Algorithm 1.
>
> ## Issue regarding the algorithm for the Frank-Wolfe step
>
> > However, one missing key point in this paper is the algorithm to compute the
> approximate gradient oracle (called ‘Frank Wolfe step’ in the paper. Though it
> is more a steepest descent step in my opinion). I looked at the appendix quite
> extensively and it seems that one should run algorithm 3 to get a $\lambda$,
> sample $x \sim \mu$ and then return $m_{\lambda}(x)$ see (99). It would be useful to have a pseudo algorithm summarizing this in the main paper.
>
> Your interpretation of this algorithm is correct and we agree that this
> algorithm is reasonably buried within the appendix. After incorporating all the
> other changes mentioned in our responses, we would like to modify the paper to
> incorporate this suggestion, subject to space requirements.
>
> ## Q3: Regarding the experiments
>
> > Are the experiments within the assumptions? It is not mentioned or justified that the functions considered are smooth and PL.
>
> For general MMD functionals, general smoothness and {\L}ojasiewicz inequalities (Assumptions 1 and 3 with $S = \mathcal{P}_2(\mathbb{R}^{d})$) are shown in [2] --- under
> the assumptions on the kernel listed in Appendix B of that paper. However, the MMD experiment in this paper (which is taken from [2]) fails to strictly satisfy the differentiability assumptions of [2] --- this is due to the ReLU terms present in the network defining the kernel. Such a technicality is likely surmountable. However, we believe that accomplishing this
> (via a study and adaption of the techniques in [2]) is
> beyond the scope of this work. Further, smoothness and PL inequalities for the functional in the Gaussian deconvolution example are, to the our best knowledge, unknown. This is despite these inequalities being known, under various assumptions, for Wasserstein-barycenter-type functionals [3].
>
> This question is instructive since it illuminates a possible source of confusion associated with the computational examples. Thus, we will add a remark to this section of the revised version which provides the following.
> - First, a concise summary of extent to which smoothness and PL inequalities are known for the computational examples.
> - Second, a clarification that the purpose of the
> computational examples is not to redo the theoretical analysis of the paper, but to demonstrate practical relevance of the proposed algorithm. The algorithm just needs a very weak of differentiability to be applied (Gateaux differentiability). We show attractive performance in applications where the assumptions required for theoretical convergence are unknown (or maybe even violated). We believe that this purpose is consistent with the pragmatic, fixed step-size adjustments to proposed method that are studied in this section of the paper.

---

> > ### Author Response · Authors · 2021-08-10
> > **Remaining response to Reviewer cwwb**
> >
> > ## Q4: Regarding the relationship to DRO
> >
> > > A significant part of the related work is dedicated to DRO? Since this work cannot handle constrained optimization (J has to be smooth) how is this work related to DRO?
> >
> > We require greater clarification from the reviewer regarding the DRO formulation to which the reviewer is referring and the role of smoothness in the context of this problem. However, the connection between (2), (15), and the DRO framework presented in [4] (also present in [5], [6, Chapter 6]) follows from the fact that (2), (15) and the problem considered in
> > [4] exhibit the same form. Indeed,  the subproblem for the FW step can be regarded as a DRO problem. As a point of clarification (we also can add a remark to the paper to reflect this), we do not claim that there is an immediate relationship between distributionally robust optimization and the problem $\min_{\mu \in \mathcal{P}_2(\mathbb{R}^{d})} J(\mu)$.
> >
> > ## Q5: The geodesically convex set $S$
> > > About the Set $S$: All your assumptions mention a set $S$: If the set $S$ is not the whole space, how do you make sure that the iterates stay in $S$? What are non-trivial examples of sets $S$?
> >
> > This question is quite informative and highlights the role of
> > Assumption 2. Perhaps it also illustrates the limitations of Assumption 2. We will update our paper with a remark which reflects the
> > following. When $S$ is not the whole set, the iterates produced by our algorithm are guaranteed to remain in $S$ by Assumption 2. This is a result of the fact that the solution of (15), computed using Proposition 1, is optimal for some Wasserstein ball of size $\widetilde{\delta} \leq \delta$-- consider first-order optimality conditions for (18). Hence, each of these iterates is guaranteed to
> > lie in $S$ (by Assumption 2) so long as $\delta \leq \Delta_2$.
> >
> > The purpose of the set $S$ is to suggest how the algorithm could be adapted when
> > considering functionals $J$ whose effective domain is a strict subset of
> > $\mathcal{P}_2(\mathbb{R}^d)$. In the Frank-Wolfe setting, perhaps the most relevant
> > example of a non-trivial $S$ is: a geodesically convex
> > subset of $\mathcal{P}_2(\mathbb{R}^d)$ given by a finite number of linear constraints. Indeed,
> > for a set of $n$ convex functions $g_i : \mathbb{R}^d \to \mathbb{R}$, and $n$ real numbers
> > $t_i \in \mathbb{R}$, define
> > $S := \\{ \mu \in \mathbb{P}_2\left( \mathbb{R}^d \right) : \int g_i \, d \mu \leq t_i
> > \\}$. Clearly, $S$ is geodesically convex and, in this context, Assumption
> > 2 would hold if $S$ is the effective domain of the functional of interest
> > $J$.
> >
> > For this case, it can be argued that Assumption 2 is brittle. However, under these linear constraints, we conjecture that (15) can be modified to incorporate $S$ with a direct extension of the duality results in [4]. This would eliminate the need for Assumption 2 and, we believe, this would be a promising line of inquiry for future work! We will add a statement to this effect in the conclusion.
> >
> > Thank you for your consideration of our response! Please let us know if you have any remaining concerns or if there is anything else that would help you!
> >
> > ## References
> >
> > [1] Boyd, Stephen, Stephen P. Boyd, and Lieven Vandenberghe. Convex optimization. Cambridge university press, 2004.
> >
> > [2] Arbel, Michael, et al. "Maximum Mean Discrepancy Gradient Flow." Advances in Neural Information Processing Systems 32 (2019): 6484-6494.
> >
> > [3] Chewi, Sinho, et al. "Gradient descent algorithms for Bures-Wasserstein barycenters." Conference on Learning Theory. PMLR, 2020.
> >
> > [4] Blanchet, Jose, and Karthyek Murthy. "Quantifying distributional model risk via optimal transport." Mathematics of Operations Research 44.2 (2019): 565-600.
> >
> > [5] Sinha, Aman, Hongseok Namkoong, and John Duchi. "Certifying Some Distributional Robustness with Principled Adversarial Training." International Conference on Learning Representations. 2018.
> >
> > [6] Kuhn, Daniel, et al. "Wasserstein distributionally robust optimization: Theory and applications in machine learning." Operations Research \& Management Science in the Age of Analytics. INFORMS, 2019. 130-166.
> >
> > [7] Migdalas, Athanasios. "A regularization of the Frank—Wolfe method and unification of certain nonlinear programming methods." Mathematical Programming 65.1 (1994): 331-345.
> >
> > [8] Luise, Giulia, et al. "Sinkhorn Barycenters with Free Support via Frank-Wolfe Algorithm." Advances in Neural Information Processing Systems 32 (2019): 9322-9333.
> >
> > [9] Garber, Dan, and Elad Hazan. "A linearly convergent variant of the conditional gradient algorithm under strong convexity, with applications to online and stochastic optimization." SIAM Journal on Optimization 26.3 (2016): 1493-1528.

---

> ### Author Response · Authors · 2021-09-01
> **Additional note to reviewer cwwb**
>
> Dear reviewer cwwb,
>
> Thank you, again, for your highly constructive comments and questions regarding our work - most importantly, thank you very much for the time that you have spent reviewing our work and responses. We recognize that duties of a referee, during the discussion period, are an order of magnitude more onerous than the duties of the authors: since we have the discretion to concentrate on the response for a single submission, while a referee will almost surely have the duty to concentrate on the response for many, different submissions.
>
> As the discussion period draws to a close, we wanted to reach out to you directly and make sure that you do not have any lingering questions or concerns that we could answer. Based upon the followup during the discussion period, we hope to have provided useful and complete responses to your questions regarding our work- particularly if these questions were brought with the objective that the ensuing discussion could materially impact your assessment of the paper. Please, do not hesitate to reach out to us if there is any additional clarification that we can provide, we are at your service.
>
> Yours truly,
> The Authors

---

### Official Review · Reviewer_yMtu · 2021-07-26

**Rating:** 9
**Confidence:** 5

**Summary:**

The paper introduces Frank-Wolfe algorithms for minimizing differentiable functionals over probability measures. It can be applied to a wide range of applications from several fields and shed light on distributionally robust optimization problems over ambiguity sets defined through the Wasserstein metric.

**Limitations And Societal Impact:**

More discussions on how to handle other types of Wasserstein metrics are necessary.

**Main Review:**

The methods and perspectives are novel. Potential applications of the proposed algorithms are wide are and remain open.

**Time Spent Reviewing:**

2

---

> ### Author Response · Authors · 2021-08-10
> **Authors' Response to yMtu**
>
> Thank you for your positive comments and appreciation of our work! In what follows, we answer the following comment pointed out in your review. In our updated version, we will provide this discussion in the conclusion and future directions section of the work.
>
> ## Extensions to Wasserstein distance for $p\neq 2$ (e.g., p= 1):
>
> > More discussions on how to handle other types of Wasserstein metrics are necessary.
>
> We will add a discussion making it more clear that the algorithm doesn't require Wasserstein differentiability to be applied (only Gateaux differentiability). But we will certainly explain that for the convergence analysis, the extension of this work to $p\neq 2$ essentially comes in two parts: 1) Extension of differentiability to $p \neq 2$ and 2) Extension of the convergence guarantees to $p \neq 2$.  In nutshell, we conjecture that the appropriate analogues of Wasserstein differentiability notion can be made in a meaningful way. However, the convergence analysis that we provide highly depends on the specific geometry of the 2-Wasserstein metric and extensions do not follow immediately from our analysis.
>
> **Extension of differentiability:** There appears to be little difficulty in extending the definition of differentiability in the paper to hold for Wasserstein distance of order $p$ and to account for functionals over $\mathcal{P}_p(\mathbb{R}^{d})$ (distributions for which the $p$-th power of the Euclidean norm is integrable).
>
> Indeed, for $p > 1$ we conjecture that Definition 1 can be extended, verbatim, since $W_p$ provides: 1) an analogous metric space structure and 2) an analogous notion of a cotangent bundle [1, Chapter 8]. The cotangent bundle has a functionally equivalent form to the $p=2$ case. That is, it is the $L^q$ closure (with $q$ conjugate to $p$) of gradients of smooth, compactly supported functions. Therefore, Definition 2 can also immediately be generalized in a meaningful way. Meanwhile, it appears that the definitions of smoothness and the {\L}ojasiewicz inequalities can be directly lifted to this setting so long as the latter inequality is stated in terms of the conjugate norm on the contangent bundle.
>
> Further, we believe that a similar extension to the $p=1$ case could be made in a meaningful way. However, this extension requires greater care than the extension for the $p>1$ case and likely necessitates the development of nontrivial results for the $W_1$ geometry that fulfil the role of Proposition 2 in Appendix A. Here, our reasoning is driven by the change in the dynamic formulation of $W_1$. Specifically, there no longer exists a characterization of absolutely continuous paths in terms of solutions to continuity equations [1, Theorem 8.3.1]. Indeed, the continuity equation no longer makes sense for $W_1$ [2, Remark 5.23] and, instead, we obtain Beckmann's minimal flow problem [2, Chapter  4] as an appropriate dynamic formulation. Using Beckmann's problem, it is possible to give meaning to the tangent and cotangent spaces of $\mathcal{P}_1(\mathbb{R}^{d})$ where the tangent space would correspond to the space of all finite, zero-mean Borel vector measures with divergence which is a scalar measure (in a weak sense) and the cotangent space would be a suitable closure (in the topological dual of the tangent space) of gradients of compactly-supported, smooth functions. Provided with this, we believe that Wasserstein differentiability (and the subsequent definitions) could be, verbatim, extended to $W_1$ in a way that is consistent, that is, in a way that does not depend on the fact that geodesics non-longer uniquely correspond to optimal transport plans.
>
> After resolving these technicalities, it would only remain to show that a suitable analogue to (28) in Proposition 2 held for the Beckmann problem and $W_1$ --- this would establish that the definitions of Wasserstein differentiability and smoothness in our paper are relevant for studying the convergence of methods which solve sequences of problems in the form of (15). However, we are currently unaware of a appropriate analogue to (28) in the literature and believe that establishing such a result may be non-trivial.
>
> **Extension of convergence guarantees:** Provided the extension of Wasserstein differentiability to $p \neq 2$ (detailed above), would the convergence guarantees of the paper still apply? Adding to this, one could ask: if the appropriate analogues of Definitions 1-4 and Assumptions 1-3 held for $\mathcal{P}_p(\mathbb{R}^{d})$ and $W_p$ with $p \neq 2$, would the results of Theorem 1 and/or Proposition 1 follow with little effort?
>
> The answer to this question is no. Specifically, the proof of Theorem 1 in Appendix B relies on the celebrated relationship between Kantorovich potentials and convex functions which is only true if $p=2$. Further, in the regime $p \neq 2$ there is a mismatch between the norms on the tangent and cotangent spaces. Thus, Lemma 2 does not immediately generalize for $p\neq 2$ without dimension dependent constants or further assumptions. It is also worth noting that the extension of Proposition 1 to $W_p$ with $p \neq 2$ is not direct. The cost $y \rightarrow \|y-x\|^p$ is not necessarily strongly convex and thus, a priori, it is not apparent that the proximal operator $\arg\min f(y) + \lambda \|y-x\|^p / 2$ can be executed without incurring in additional computational costs.
>
> Fortunately, we do not believe that these issues are reasons for pessimism - as regards the extension of this work to $W_p$ for $p \neq 2$. Rather, these issues demonstrate that such an extension would be interesting. It is reasonable to expect that a suitable use of $c$-concavity [2, Section 1.6] for $c(x,y) = \|x-y\|^p$, coupled with ideas from high-order, proximal-point methods [3], could address the non-trivial issues discussed in the previous paragraphs.
>
> ## References
> [1] Ambrosio, Luigi, Nicola Gigli, and Giuseppe Savaré. Gradient flows: in metric spaces and in the space of probability measures. Springer Science \& Business Media, 2008.
>
> [2] Santambrogio, Filippo. Optimal Transport for Applied Mathematicians: Calculus of Variations, PDEs, and Modeling. Vol. 87. Birkhäuser, 2015.
>
> [3] Nesterov, Jurij Evgenevic. Inexact accelerated high-order proximal-point methods. No. UCL-Université Catholique de Louvain. CORE, 2020.

---

### Official Review · Reviewer_8BEc · 2021-07-28

**Rating:** 8
**Confidence:** 3

**Summary:**

This work investigates optimization in infinite dimensional functionals of probability measures. The authors first define a notion of gradient of functional using Wasserstein gradient flows. Then, the authors propose a Frank-Wolfe procedure in which gradient steps can be computed using finite dimensional and convex optimization methods. The authors show that their algorithm is guaranteed to converge under some assumptions by choosing proper step sizes. Finally, the authors apply their method to two non-parametric estimation problems.

**Limitations And Societal Impact:**

The original Frank-Wolfe algorithm proposed by the authors might be computationally expensive for high-dimensional cases as it need a large number of ascent step for $\lambda$ in computation of gradient steps. To address this issue, the authors provide a uniform strategy for $\lambda$  which can reduce computational costs. The authors claim that they do not foresee any negative societal impact as the contributions of their work are theoretical.

**Main Review:**

The paper aims to optimize a functional of probability measures. This problem is very important and has a lot of applications in machine learning. The paper is well written and has a nice logical flow. The claims in the paper are also well supported. The notion of Wasserstein differentiability introduced to define the gradient of functional is novel, and the proposed Frank-Wolfe procedure is promising. In particular, I like their “primal-dual” algorithm for solving the Frank-Wolfe step which can implicitly maintain a primal-feasible iterate and also make progress on the primal-dual gap.

Some questions:

(1) The notion of Wasserstein differentiability is defined with respect to Wasserstein distance of order 2. I am curious whether this definition can be easily extended to $p$-order Wasserstein distance (e.g., p=1). Similarly, if considering 1-Wasserstein distance, does the algorithm (also the convergence guarantees) still apply?

(2) In Definition 1, the authors define the gradient of a functional in terms of cotangent bundle. If I understand correctly, the element in cotangent bundle should be a mapping from $\mathbb{R}^d\rightarrow \mathbb{R}^d$. Then in Section 4.2, the authors claim that the Wasserstein derivative of squared MMD is the difference between the mean embedding of $\mu$ and $\nu$, i.e., $f_{\mu}^*(\cdot)$. However $f_{\mu}^*(\cdot)$ is a mapping from $\mathbb{R}^m\rightarrow \mathbb{R}$. Isn't this contradictory? Can the authors clarify this?

(3) All assumptions and also Theorem 1 assume a geodesically convex set S. I am interested in what S would be like in practice. Is there any functional of probability measures such that S would be the whole space $\mathcal{P}_2(\mathbb{R}^d)$? Moreover, the authors claim that the Frank-Wolfe problem (15) exhibits a dual form given by (18). Can the authors please provide some references for this dual result?

Minor comments:

(1) The authors should comply with the NeurIPS 2021 style where there is no "Broader impact" Section and the checklist should be after the References Section.

(2) There are some typos, e.g.,
- In Line 108, "a selections"-> "a selection"
- In Line 116, "a aesthetic"-> "an aesthetic"
- In Line 260, $\nu$ should be $\mu$
- In line 163 and (15), $W$ should be $\mathcal{W}$.
- What is $\bar{\mathbb{R}}$?

**Time Spent Reviewing:**

20

---

> ### Author Response · Authors · 2021-08-10
> **Author's Response to Reviewer 8BEc**
>
> We appreciate and thank the reviewer for their positive comments, especially concerning the the proposed Frank-wolfe procedure and its interesting convergence analysis via Wasserstein geometry. We would also like to thank the reviewer for their thoughtful and constructive feedback on our work. We now provide responses to each of the important questions you have raised.
>
> ## Q1: Extensions to Wasserstein distance for $p\neq 2$ (e.g., p= 1):
>
> > (1) The notion of Wasserstein differentiability is defined with respect to Wasserstein distance of order 2. I am curious whether this definition can be easily extended to $p$-order Wasserstein distance (e.g., $p=1$). Similarly, if considering 1-Wasserstein distance, does the algorithm (also the convergence guarantees) still apply?
>
> Thanks for pointing out a potential extension of the work that we believe to be promising. First, it is important to note that to actually implement the algorithm we do *not* need the function to be Wasserstein differentiable, but only Gateaux differentiable (the Gateaux derivative is what is known as the influence function as we mention in the paper). However, our proof of convergence requires basically the Gateaux derivative to be a differentiable function itself, this is basically the 2-Wasserstein differentiability requirement. Now, regarding the extension of the proof of convergence development, the answer to this question comes in essentially two parts: 1) Extension of differentiability to $p \neq 2$ and 2) Extension of the convergence guarantees to $p \neq 2$.  In nutshell, we conjecture that the appropriate analogues of Wasserstein differentiability notion can be made in a meaningful way. However, the convergence analysis part is hard to extend, which highly depends on the specific structure of 2-wasserstein metric.
>
>  **Extension of differentiability:** There appears to be little difficulty in extending the definition of differentiability in the paper to hold for Wasserstein distance of order $p$ and to account for functionals over $\mathcal{P}_p(\mathbb{R}^{d})$ (distributions for which the $p$-th power of the Euclidean norm is integrable).
>
> Indeed, for $p > 1$ we conjecture that Definition 1 can be extended, verbatim, since $W_p$ provides: 1) an analogous metric space structure and 2) an analogous notion of a cotangent bundle [1, Chapter 8]. The cotangent bundle has a functionally equivalent form to the $p=2$ case. That is, it is the $L^q$ closure (with $q$ conjugate to $p$) of gradients of smooth, compactly supported functions. Therefore, Definition 2 can also immediately be generalized in a meaningful way. Meanwhile, it appears that the definitions of smoothness and the {\L}ojasiewicz inequalities can be directly lifted to this setting so long as the latter inequality is stated in terms of the conjugate norm on the co-tangent bundle.
>
> Further, we believe that a similar extension to the $p=1$ case could be made in a meaningful way. However, this extension requires greater care than the extension for the $p>1$ case and likely necessitates the development of nontrivial results for the $W_1$ geometry that fulfil the role of Proposition 2 in Appendix A. Here, our reasoning is driven by the change in the dynamic formulation of $W_1$. Specifically, there no longer exists a characterization of absolutely continuous paths in terms of solutions to continuity equations [1, Theorem 8.3.1]. Indeed, the continuity equation no longer makes sense for $W_1$ [2, Remark 5.23] and, instead, we obtain Beckmann's minimal flow problem [2, Chapter  4] as an appropriate dynamic formulation. Using Beckmann's problem, it is possible to give meaning to the tangent and cotangent spaces of $\mathcal{P}_1(\mathbb{R}^{d})$ where the tangent space would correspond to the space of all finite, zero-mean Borel vector measures with divergence which is a scalar measure (in a weak sense) and the cotangent space would be a suitable closure (in the topological dual of the tangent space) of gradients of compactly-supported, smooth functions. Provided with this, we believe that Wasserstein differentiability (and the subsequent definitions) could be, verbatim, extended to $W_1$ in a way that is consistent, that is, in a way that does not depend on the fact that geodesics non-longer uniquely correspond to optimal transport plans.
>
> After resolving these technicalities, it would only remain to show that a suitable analogue to (28) in Proposition 2 held for the Beckmann problem and $W_1$ - this would establish that the definitions of Wasserstein differentiability and smoothness in our paper are relevant for studying the convergence of methods which solve sequences of problems in the form of (15). However, we are currently unaware of a appropriate analogue to (28) in the literature and believe that establishing such a result is not immediate.
>
>
> **Extension of convergence guarantees:** Provided the extension of Wasserstein differentiability to $p \neq 2$ (detailed above), would the convergence guarantees of the paper still apply? Adding to this, one could ask: if the appropriate analogues of Definitions 1-4 and Assumptions 1-3 held for $\mathcal{P}_p(\mathbb{R}^{d})$ and $W_p$ with $p \neq 2$, would the results of Theorem 1 and/or Proposition 1 follow with little effort?
>
> The answer to this question is no. Specifically, the proof of Theorem 1 in Appendix B relies on the celebrated relationship between Kantorovich potentials and convex functions which is only true if $p=2$. Further, in the regime $p \neq 2$ there is a mismatch between the norms on the tangent and cotangent spaces. Thus, Lemma 2 does not immediately generalize for $p\neq 2$ without dimension dependent constants or further assumptions. It is also worth noting that the extension of Proposition 1 to $W_p$ with $p \neq 2$ is non-trivial. The cost $y \rightarrow \|y-x\|^p$ is not necessarily strongly convex and thus, a priori, it is not apparent that the proximal operator $\arg\min f(y) + \lambda \|y-x\|^p / 2$ can be made computable.
>
> Fortunately, we do not believe that these issues are reasons for pessimism - as regards the extension of this work to $W_p$ for $p \neq 2$. Rather, these issues demonstrate that such an extension would be interesting. It is reasonable to expect that a suitable use of $c$-concavity [2, Section 1.6] for $c(x,y) = \|x-y\|^p$, coupled with ideas from high-order, proximal-point methods [3], could address the non-trivial issues discussed in the previous paragraphs.
>
> ## Q2: Wasserstein derivative of squared MMD
>
> > (2) In Definition 1, the authors define the gradient of a functional in terms
> of cotangent bundle. If I understand correctly, the element in cotangent bundle
> should be a mapping from $\mathbb{R}^d \to \mathbb{R}^d$. Then in Section 4.2, the authors claim
> that the Wasserstein derivative of squared MMD is the difference between the
> mean embedding of $\mu$ and $\nu$, i.e., $f_\mu^*$. However $f_\mu^*$ is a
> mapping from $\mathbb{R}^m \to \mathbb{R}$. Isn't this contradictory? Can the authors clarify
> this?
>
> Thank you for this observation. We will update the paper to clarify that
> $\nabla f_\mu^* : \mathbb{R}^d \to \mathbb{R}^d$ (such that it exists) provides the Wasserstein derivative for the squared MMD. The function $f_\mu^*$ is more precisely the Gateaux differential (or \textit{first variation}) of squared MMD [4] (also known as the influence function). As discussed in Remark 1, the Wasserstein derivative is, under sufficient regularity, the gradient of the Gateaux differential - this is essentially a consequence of the fact that geodesics in $\mathcal{W}$ satisfy a corresponding continuity equation (Appendix A). We refer to Remark 1 and the references therein for further details.
>
> ## Q3: The geodesically convex set $S$
>
> > (3) All assumptions and also Theorem 1 assume a geodesically convex set $S$. I am interested in what S would be like in practice. Is there any functional of probability measures such that $S$ would be the whole space $\mathcal{P}(\mathbb{R}^d)$.
>
> This question is quite instructive and we will update the paper with a short remark in the spirit of the following answer which should help illustrate the use of the set $S$. In practice, $S$ can be any geodecially convex set in $\mathcal{P}_2(\mathbb{R}^{d})$. The simplest example is the whole space $\mathcal{P}_2(\mathbb{R}^{d})$, e.g., minimizing Wasserstein distance or a Wasserstein barycenter functional. Besides that, slightly less-trivial example would be a geodescially convex
> subset of $\mathcal{P}_2(\mathbb{R}^d)$ given by a finite number of linear constraints. Indeed,
> for a set of $n$ convex functions $g_i : \mathbb{R}^d \to \mathbb{R}$, and $n$ real numbers
> $t_i \in \mathbb{R}$, define $S := \\{ \mu \in \mathbb{P}_2\left( \mathbb{R}^d \right) : \int g_i \, d \mu \leq t_i
> \\}$.
>
> ## Q4: The reference that supports the FW problem (15) exhibits a dual form:
>
> > Moreover, the authors claim that the Frank-Wolfe problem (15) exhibits a dual form given by (18). Can the authors please provide some references for this dual result?
>
> Thank you for spotting this! We refer reviewer to [5, Theorem 1] and will add this reference to the appropriate section in the paper!

---

> > ### Author Response · Authors · 2021-08-10
> > **References for response**
> >
> > [1] Ambrosio, Luigi, Nicola Gigli, and Giuseppe Savaré. Gradient flows: in metric spaces and in the space of probability measures. Springer Science \& Business Media, 2008.
> >
> > [2] Santambrogio, Filippo. Optimal Transport for Applied Mathematicians: Calculus of Variations, PDEs, and Modeling. Vol. 87. Birkhäuser, 2015.
> >
> > [3] Nesterov, Jurij Evgenevic. Inexact accelerated high-order proximal-point methods. No. UCL-Université Catholique de Louvain. CORE, 2020.
> >
> > [4] Arbel, Michael, et al. "Maximum Mean Discrepancy Gradient Flow." Advances in Neural Information Processing Systems 32 (2019): 6484-6494.
> >
> > [5] Blanchet, Jose, and Karthyek Murthy. "Quantifying distributional model risk via optimal transport." Mathematics of Operations Research 44.2 (2019): 565-600.

---

### Author Response · Authors · 2021-08-28
**Further discussion**

Dear Reviewers,

Many thanks once again for your time and effort in reviewing our paper. We really appreciate your feedback.

We hope that we have understood and addressed all of your concerns about our submission. In case there are any remaining issues we would be glad to address them.


Best regards,

The Authors.

---

### Decision · Program_Chairs · 2021-09-27

**Decision:**

Accept (Poster)

**Comment:**

I carefully read the interactions between the reviewers and the authors as initially there was some strong split between the reviewers. Some of the things have been clarified in the meantime, so that the overall assessment seems to be that this paper is a good paper. Certain issues regarding notions and review of related methods etc have been raised however. I would like the authors to give the reviews appropriate considerations when preparing their revision. The authors answers have been taking into consideration.